# $f$-Divergence Regularized RLHF:
# Two Tales of Sampling and Unified Analyses

**Di Wu** [1]  **Chengshuai Shi** [2]  **Jing Yang** [1]  **Cong Shen** [1]

## Abstract

Reinforcement Learning from Human Feedback (RLHF) has become a cornerstone technique for post-training large language models. While most existing approaches rely on the reverse KL-regularization, recent empirical studies have begun exploring alternative divergences (e.g., forward KL, chi-squared) as regularizers in RLHF. However, a unified theoretical understanding of general $f$-divergence regularization remains under-explored. To fill this gap, this work develops a comprehensive theoretical framework for online RLHF with a general $f$-divergence regularized objective. Rather than treating each possible divergence function individually, we adopt a holistic perspective across the entire function class and propose two algorithms based on distinct sampling principles. The first extends the classical optimism principle with a carefully designed exploration bonus, while the second introduces a new method that exploits the sensitivity of the optimal policy to reward perturbations under $f$-divergence regularization. Theoretical analysis shows that $O(\log T)$ regret and $O(1/T)$ sub-optimality gap are achievable, establishing provable efficiency of both algorithms and, to the best of our knowledge, the first performance bounds for online RLHF under general $f$-divergence regularization.

## 1. Introduction

In recent years, Reinforcement Learning from Human Feedback (RLHF) has emerged as a cornerstone of the post-training stage for large language models (LLMs) (Bai et al., 2022; Ouyang et al., 2022; Rafailov et al., 2023; Dong et al., 2023; Zhao et al., 2023). It has been instrumental in enabling the alignment and deployment of today's most widely used LLMs (Achiam et al., 2023; Bai et al., 2022; Touvron et al., 2023; Team et al., 2023). Compared with canonical reinforcement learning (RL) (Sutton et al., 2018), RLHF is typically formulated with a Kullback–Leibler (KL) divergence term between the learned policy and a reference policy, a setup often referred to as *KL-regularized RL*. In the most common single-turn setting, this formulation reduces to KL-regularized contextual bandits (CB). Despite the additional analytical challenges introduced by KL regularization, recent research has established sharp theoretical guarantees: Zhao et al. (2025a) show that KL-regularized RL/CB achieves logarithmic regret $O(\log T)$ in the online setting and Zhao et al. (2025b) prove a sample complexity $O(\varepsilon^{-1})$ in the offline setting for $\varepsilon$-optimality under single-policy coverage.

Despite its broad adoption in RLHF post-training, KL divergence does not yield a sufficiently robust training process. For instance, under KL regularization, *over-optimization* (Huang et al., 2025; Rafailov et al., 2024) can arise because sparse offline data forces the reward model to generalize out of distribution; the learned policy then moves into these regions and exploits reward-model errors, yielding high proxy scores but poorer real performance. To address this challenge, different types of $f$-divergence have been adopted as regularizers to improve the training behavior, and some recent results already demonstrate distinctive advantages in RL and preference optimization. Huang et al. (2025) propose to use mixed chi-squared divergence, and demonstrate it as a stronger regularizer that mitigates over-optimization and improves robustness compared to reverse KL. Forward KL regularization has been studied in Shan et al. (2025), which mitigates out-of-distribution issues during the alignment process of diffusion models. In addition, broader divergence families such as $\alpha$-divergences provide flexible trade-offs between exploration and exploitation (Belousov & Peters, 2018). Together, these developments highlight the potential benefits of divergences beyond KL divergence and motivate a systematic study of $f$-divergence regularized RLHF, where KL divergence is a special case within a more general and versatile family.

---

[1]Department of Electrical and Computer Engineering, University of Virginia, Virginia, United States [2]Princeton Language and Intelligence, Princeton University, New Jersey, United States. Correspondence to: Cong Shen <cong@virginia.edu>.

*Proceedings of the $43^{rd}$ International Conference on Machine Learning*, Seoul, South Korea. PMLR 306, 2026. Copyright 2026 by the author(s).

Although many studies have established favorable results for different types of regularization, unified analyses for the general $f$-divergence regularized RLHF are still lacking. Aminian et al. (2026) focus on forward and reverse KL divergences, but do not explore other divergence functions. Go et al. (2023); Han et al. (2024) have studied the $f$-divergence alignment, but they focus on a different $f$-divergence metric than our paper. Wang et al. (2024); Sun et al. (2025) apply the canonical $f$-divergence regularization term and discuss $f$-divergence Direct Preference Optimization (DPO) as a variant of RLHF. Their study, however, is largely empirical without theoretical performance guarantees. Recently, Zhao et al. (2025b) theoretically analyze the general $f$-divergence regularized RLHF. However, their work is limited to the offline setting, leaving the online setting unexplored. Our paper fills this important gap, and the main contributions can be summarized as follows:

• We propose two exploration strategies for the $f$-divergence regularized RLHF. The first approach, **optimism-based exploration**, addresses reward estimate errors through the celebrated *optimism in the face of uncertainty* principle. The second approach, **derivative-based exploration**, leverages the sensitivity of the function via employing derivative-based signals for efficient exploration.

• We establish a novel theoretical insight applicable to any $f$-divergence: the derivative of $f$ can be interpreted as part of the uncertainty. This interpretation directly yields the principled derivative-based sampling policy, and plays an important role in our theoretical analysis. This **"derivative-as-uncertainty"** intuition is fundamental in the sense that it is applicable to any $f$-divergence RLHF, or even general machine learning that leverages $f$-divergence. To the best of our knowledge, this is the first paper to interpret the derivative of $f$ as a key uncertainty factor in promoting exploration.

• Our theoretical analysis establishes a regret upper bound of $O(\log(T))$ over a time horizon of $T$ for the optimism-based exploration algorithm. Furthermore, we prove an $O(1/T)$ sub-optimality gap for derivative-based exploration. To the best of our knowledge, this is the first time that $O(\log(T))$ regret and $O(1/T)$ sub-optimality gap are attained within the online $f$-divergence regularized RLHF framework.

## 2. $f$-divergence Regularized RLHF

Following the theoretical RLHF framework established in Xiong et al. (2023); Zhu et al. (2023); Ye et al. (2024), we model the language model (LM) as a policy $\pi : \mathcal{X} \to \mathcal{A}$, which takes a context $x \in \mathcal{X}$ (i.e., a prompt) as input and outputs an action $a \in \mathcal{A}$ (i.e., a response). A key characteristic of RLHF is that the learning agent receives preference feedback over actions (typically action pairs)

instead of direct reward feedback. Following the canonical RLHF framework (Ziegler et al., 2019; Ouyang et al., 2022; Bai et al., 2022), this work focuses on the Bradley-Terry (BT) model (Bradley & Terry, 1952) to capture the pairwise preferences, as given in Definition 2.1. To ease the notation, given a context $x$ and an action pair $(a^1, a^2)$, we denote that the learning agent receives a binary feedback $y \in \{0, 1\}$, with $y = 0$ meaning $a^1$ is preferred over $a^2$ (i.e., $a^1 \succ a^2$) and $y = 0$ meaning $a^2$ is preferred over $a^1$ (i.e., $a^1 \prec a^2$).

**Definition 2.1** (Bradley-Terry Model). The probability of $a^1$ being preferred over $a^2$ under context $x$ is modeled as

$$\mathbb{P}(a^1 \succ a^2 | x, a^1, a^2) = \frac{\exp(r^*(x, a^1))}{\exp(r^*(x, a^1)) + \exp(r^*(x, a^2))}$$
$$= \sigma\big(r^*(x, a^1) - r^*(x, a^2)\big), \tag{1}$$

where $r^* : \mathcal{X} \times \mathcal{A} \to [0, 1]$ is a reward function that captures the performance of one action $a \in \mathcal{A}$ under a context $x \in \mathcal{X}$, and $\sigma(\cdot)$ denotes the sigmoid function.

### 2.1. Learning Objective with $f$-divergence Regularization

During the RLHF process, it is common to regularize the finetuned model to stay "close" to a reference model, denoted as policy $\pi_0$, which is typically the checkpoint after supervised finetuning. In most practical implementations, the following regularized learning objective is considered, with the reverse KL divergence as the regularizer:

$$J_{\mathrm{KL}}(\pi) = \mathbb{E}_{x \sim \rho} \mathbb{E}_{a \sim \pi(\cdot|x)} \big[ r^*(x, a) - \eta^{-1} D_{\mathrm{KL}}(\pi, \pi_0 | x) \big], \tag{2}$$

where $D_{\mathrm{KL}}(\pi, \pi_0 | x) := D_{\mathrm{KL}}(\pi(\cdot|x) || \pi_0(\cdot|x))$ and $\rho$ denotes the context distribution, i.e., $x$ is independently sampled from $\rho$. The existing theoretical investigations on RLHF (Xiong et al., 2023; Ye et al., 2024; Zhao et al., 2024; 2025a;b) have been mostly focused on this objective.

As mentioned in Section 1, some recent empirical efforts have explored leveraging alternative divergences as regularizers in the RLHF learning objective, such as Huang et al. (2025); Wang et al. (2024), where various benefits have been discovered. Motivated by these empirical advances, this work studies the theoretical properties of RLHF beyond reverse KL-regularization. Specifically, instead of studying each divergence choice separately (i.e., a specific function $f$), we take a holistic perspective and study a general function class, i.e., the $f$-divergence, defined in the following.

**Definition 2.2** ($f$-divergence). For any convex function $f : \mathbb{R}^+ \to \mathbb{R}$ with $f(1) = 0$ and strictly convex around 1, we define a divergence of two distributions $p$ and $q$ relative to $f$ as

$$D_f(p||q) = \mathbb{E}_{q(x)} \left[ f\left( \frac{p(x)}{q(x)} \right) \right], \tag{3}$$

which is commonly referred to as the $f$-*divergence*.

The $f$-divergence covers a broad class of widely used divergences, including forward KL divergence, reverse KL divergence, Jensen-Shannon (JS) divergence, and total variation distance, etc., by choosing the specific function $f$. Some details on the $f$-divergence can be found in Table 1 and Appendix B.

With $f$-divergence and its generality introduced, the RLHF learning objective under regularization can be formulated as

$$J_f(\pi) = \mathbb{E}_{x\sim\rho}\mathbb{E}_{a\sim\pi(\cdot|x)}\left[r^*(x,a) - \eta^{-1}D_f(\pi,\pi_0|x)\right], \quad (4)$$

where $D_f(\pi,\pi_0|x) := D_f(\pi(\cdot|x)||\pi_0(\cdot|x))$. It can be observed that this objective is more general, and recovers the objectives regularized with specific divergences, such as Equation (2) and the ones adopted in Huang et al. (2025); Wang et al. (2024), as special cases.

Under this target, the optimal policy $\pi_f^*$ and the value suboptimality gap of policy $\pi$ with respect to $\pi_f^*$ can be defined, respectively, as

$$\pi_f^* := \arg\max_\pi J_f(\pi), \quad (5)$$

$$\texttt{SubOpt}_f(\pi) := J_f(\pi_f^*) - J_f(\pi), \quad (6)$$

where the latter will be used in subsequent sections to measure the quality of a learned policy.

### 2.2. Optimal Policy under $f$-divergence Regularization

In the following, we discuss the optimal policy for the $f$-divergence regularized RLHF, i.e., $\pi_f^*$ in Equation (5), which turns out to admit a closed-form solution under some restrictions on $f$.

**Proposition 2.3.** *If $\pi_0(a|x) > 0$ for any possible $x$ and $a$, and $f'$ is invertible with $0 \notin \mathrm{dom}(f')$ where $\mathrm{dom}(f')$ denotes the domain of function $f'$, the optimal policy $\pi_f^*$ defined in Equation (5) can be expressed as*

$$\pi_f^*(a|x) = \pi_0(a|x)f'^{-1}\left(\eta(r^*(x,a) - \lambda_f^*(x))\right),$$

*where for each $x$, $\lambda_f^*(x)$ is a constant satisfying $\sum_{a\in\mathcal{A}}\pi_0(a|x)f'^{-1}(\eta(r^*(x,a)-\lambda_f^*(x))) = 1$, i.e., making $\pi_f^*(\cdot|x)$ a valid distribution.*

Although it is not as obvious as in the KL divergence case (Rafailov et al., 2023), the corresponding optimal policy in Proposition 2.3 remains the same for rewards with differences that only depend on $x$. Specifically, if two reward functions satisfy that $r_1(a,x) - r_2(a,x)$ depends on only $x$ but not $a$, then their induced optimal policies are the same. This is an important property leveraged in the subsequent analyses, and we have more discussions on the optimal policy in Appendix B.

The remainder of this paper will consider the condition in Proposition 2.3, which is: $\pi_0(a|x) > 0$ for any valid $x$ and $f'$ is invertible with $0 \notin \mathrm{dom}(f')$. This class of functions $f$ allows Equation (5) to have an explicit solution (c.f. Proposition 2.3), and still covers a lot of interesting divergences, such as the reverse KL divergence, JS divergence, and forward KL.

**Other Notations.** To facilitate presentation in later presentations, we consider a reward class $\mathcal{R}_\Theta := \{r_\theta : \theta \in \Theta\}$ which is parameterized by class $\Theta$. For each reward function $r$, its corresponding optimal policy $\pi_{f,r}$ under a $f$-divergence regularization is defined as $\pi_{f,r} := \arg\max_\pi \mathbb{E}_{x\sim\rho,a\sim\pi(\cdot|x)}[r(x,a) - \eta^{-1}D_f(\pi,\pi_0|x)]$, which can be further expressed as $\pi_{f,r}(a|x) = \pi_0(a|x)f'^{-1}(\eta(r(x,a) - \lambda_{f,r}(x)))$ according to Proposition 2.3. When there is no ambiguity, abbreviations $\pi_r := \pi_{f,r}$, $\pi_\theta := \pi_{f,r_\theta}$, $\lambda_r := \lambda_{f,r}$ and $\lambda_\theta := \lambda_{f,r_\theta}$ are often adopted. We further denote $h := (f')^{-1}$ as the inverse function of the derivative $f'$.

## 3. Optimism-based Exploration

### 3.1. Algorithm Design

In this section, we develop an optimism-based algorithm, presented in Algorithm 1, that takes into account the uncertainty of online data via carefully constructed confidence bounds. This is a novel extension of previous optimism-based designs in the theoretical RLHF studies, such as Xiong et al. (2023); Ye et al. (2024); Zhao et al. (2025a), beyond their original considerations of KL-regularization.

For Algorithm 1, in each iteration $t$, actions $a_t^1$ is drawn from the current policy $\pi_t$ and $a_t^2$ from the previous policy $\pi_{t-1}$ (Step 4). After observing the preference signal $y_t$, the maximum likelihood estimation (MLE) is performed to obtain an estimated $\theta_t$ based on all the previously collected data (Step 6). Then, a bonus term $b_t$ (see the last term in Line 7) is added to the reward $r_{\theta_t}$ to form an optimistic estimate $\hat{r}_t$ for the purpose of optimism, where the explicit form of $b_t$ is given in Theorem 3.3. Finally, an updated policy $\pi_{t+1}$ is obtained as the optimal policy corresponding to the optimistic reward estimate $\hat{r}_t$.

### 3.2. Theoretical Analysis

The performance of Algorithm 1 for online RLHF with $f$-divergence regularization is characterized via a newly derived regret bound. The standard realizability assumption is first introduced, which indicates that the true reward model $r^*$ is in the considered reward class $\mathcal{R}_\Theta$.

**Assumption 3.1.** It is assumed that $\mathcal{R}_\Theta$ is finite, i.e., $N_\mathcal{R} = |\mathcal{R}_\Theta| < \infty$, and realizable, i.e., there exists $\theta^* \in \Theta$ that $r_{\theta^*}$ is the true reward function $r^*$.

*Table 1.* Summary of some commonly used $f$-divergences including their derivatives.

| $f$-divergence | $f(x)$ | $f'(x)$ | $0 \notin$ Domain of $f'(x)$ |
|---|---|---|---|
| Reverse KL | $x \log x$ | $\log x + 1$ | ✓ |
| Forward KL | $-\log x$ | $-\frac{1}{x}$ | ✓ |
| JS-divergence | $x \log x - (x+1)\log((x+1)/2)$ | $\log(2x/(1+x))$ | ✓ |
| Chi-squared KL | $x \log x + (x-1)^2$ | $\log x + 2x - 1$ | ✓ |
| Total Variation | $\frac{1}{2}|x-1|$ | $x > 1 ? \frac{1}{2} : -\frac{1}{2}$ | ✗ |
| Chi-squared | $(x-1)^2$ | $2(x-1)$ | ✗ |

---

**Algorithm 1** Optimism-based Exploration under $f$-divergence Regularized RLHF

---

1: **Input:** parameter $\eta$, reference policy $\pi_0$, reward function class $\mathcal{R}_\Theta = \{r_\theta | \theta \in \Theta\}$
2: Initialize $\pi_t \leftarrow \pi_0$
3: **for** $t = 1, \ldots, T$ **do**
4:  Sample context $x_t \sim \rho$ and two actions $a_t^1 \sim \pi_t(\cdot|x_t), a_t^2 \sim \pi_{t-1}(\cdot|x_t)$
5:  Observe preference label $y_t \in \{0, 1\}$
6:  Perform the maximum likelihood estimation with $\{(x_i, a_i^1, a_i^2, y_i)\}_{i=1}^t$:

$$\theta_t \leftarrow \arg\max_{\theta \in \Theta} \sum_{i \in [t]} \Big( y_i \cdot \log \sigma(r_\theta(x, a_i^1) - r_\theta(x, a_i^2))$$
$$+ (1 - y_i) \cdot \log \sigma(r_\theta(x, a_i^2) - r_\theta(x, a_i^1)) \Big)$$

7:  Construct the optimistic reward $\hat{r}_t(\cdot, \cdot) = r_{\theta_t}(\cdot, \cdot) + \mathbb{E}_{a \sim \pi_t} b_t(\cdot, \cdot, a)$ according to Theorem 3.3
8:  Update $\pi_{t+1}$ as the optimal policy corresponding to $\hat{r}_t(\cdot, \cdot)$
9: **end for**

---

Before presenting the main theorem of this section, we first give the following definition of Eluder dimension, which is commonly adopted for measuring the effect of out-of-sample data, e.g., in Zhao et al. (2025a).

**Definition 3.2** (Eluder Dimension). Under the Bradley-Terry model, for any sequence $D_{t-1} = \{(x_i, a_i^1, a_i^2)\}_{i=1}^{t-1}$, we define the uncertainty of $(x, a^1, a^2)$ with respect to $\mathcal{R}_\Theta$ as:

$$U(\xi, x, a^1, a^2; \mathcal{R}_\Theta, \mathcal{D}_{t-1})$$
$$= \sup_{R_1, R_2 \in \mathcal{R}_\Theta} \frac{\Delta R(x, a^1, a^2)}{\sqrt{\xi + \sum_{i=1}^{t-1}(\Delta R(x, a_i^1, a_i^2))^2}}, \quad (7)$$

where $\Delta R(x, a^1, a^2) := R_1(x, a^1) - R_1(x, a^2) - R_2(x, a^1) + R_2(x, a^2)$. The corresponding Eluder dimen-

sion is

$$d(\mathcal{R}_\Theta, \xi, T)$$
$$:= \sup_{x_{1:T}, a_{1:t}^1, a_{1:T}^2} \sum_{t \in [T]} \min\{1, [U(\xi, x, a_t^1, a_t^2; \mathcal{R}_\Theta, \mathcal{D}_{t-1})]^2\}.$$

**Theorem 3.3.** *Under Assumption 3.1 and the conditions in Proposition 2.3, for any $\delta > 0$, taking $\beta_T^2 = 4e \log(N_\mathcal{R} T/\delta)$ and*

$$b_t(x, a^1, a^2) = \min\{1, \beta_T U(\xi, x, a^1, a^2; \mathcal{R}_t, \mathcal{D}_t)\},$$

*with probability at least $1 - \delta$, the cumulative regret of Algorithm 1 satisfies that*

$$\text{Regret}_f(T) := \sum_{t \in [T]} \text{SubOpt}_f(\pi_t)$$
$$= \mathcal{O}\left(\eta \mathcal{C}(f, \mathcal{R}_\Theta, \eta) \log(N_\mathcal{R} T/\delta) d(\mathcal{R}_\Theta, \xi, T)\right), \quad (8)$$

*where*

$$\mathcal{C}(f, \mathcal{R}_\Theta, \eta) = \max_{r \in \overline{\mathcal{R}_\Theta}} \max_{x, a} \frac{h'(\eta(r(x, a)) - \lambda_r(x))}{h(\eta(r(x, a)) - \lambda_r(x))},$$

*and $\overline{\mathcal{R}_\Theta}$ is the convex hull of $\mathcal{R}_\Theta$.*

*Remark* 3.4. It can be observed that Theorem 3.3 establishes a logarithmic regret bound for Algorithm 1 *regardless of the choice of $f$*. Notably, for the general $f$-divergence, it has the same regret order as that achieved by Zhao et al. (2025a) under the reverse KL divergence, which further validates the efficiency of the proposed design. The constant $\mathcal{C}(f, \mathcal{R}_\Theta, \eta)$ measures the difference between $h$ and $h'$ where $h = (f')^{-1}$. Intuitively, this constant measures how strongly the optimal policy changes when the reward estimate is perturbed. When this ratio is large, a small reward estimation error could produce a relatively large change in the induced policy, and the regret bound becomes correspondingly larger. Therefore, $\mathcal{C}(f, \mathcal{R}_\Theta, \eta)$ is the term through which the choice of $f$-divergence enters the regret analysis and affects the regret bound. We will further discussed in Section 5.

*Remark* 3.5. Regarding the Eluder dimension constant $d(\mathcal{R}_\Theta, \xi, T)$, it is affected by the structure of $\mathcal{R}_\Theta$. If the reward function class $\mathcal{R}_\Theta$ is linear, e.g., $\mathcal{R}_\Theta = \{\theta^\top \phi(\cdot, \cdot) :$

$\theta \in \mathbb{R}^d, \|\theta\|_2 \leq B\}$, and let the covariance matrix be $\Sigma_t = \sum_t \xi/B \cdot I + (\phi(x_i, a_i^1) - \phi(x_i, a_i^2))(\phi(x_i, a_i^1) - \phi(x_i, a_i^2))^\top$, then the uncertainty factor can be bounded as

$$
\begin{aligned}
&U(\xi, x, a^1, a^2; \mathcal{R}_\Theta, \mathcal{D}) \\
&= \sup_{\theta_1, \theta_2} \frac{(\theta_1 - \theta_2)^\top (\phi(x_i, a_i^1) - \phi(x_i, a_i^2))}{\sqrt{\xi + \sum_{t=1}^i ((\theta_1 - \theta_2)^\top (\phi(x_i, a_i^1) - \phi(x_i, a_i^2)))^2}} \\
&\leq \sup_{\theta_1, \theta_2} \frac{(\theta_1 - \theta_2)^\top (\phi(x_i, a_i^1) - \phi(x_i, a_i^2))}{\sqrt{(\theta_1 - \theta_2)^\top \Sigma_i (\theta_1 - \theta_2)}} \\
&\leq \|\phi(x_i, a_i^1) - \phi(x_i, a_i^2)\|_{\Sigma_i}.
\end{aligned}
$$

Furthermore, $d(R, \xi, T) \leq \sum \|\phi(x_i, a_i^1) - \phi(x_i, a_i^2)\|_{\Sigma_i}$ can be bounded as $O(\log T)$ using the classical elliptical potential lemma (Carpentier et al., 2020).

**Proof sketch:** To derive Equation (8), we first leverage the closed-form solution of $\pi_\theta$ in Proposition 2.3 to obtain the derivative of $\pi_\theta, \lambda_\theta$ to $r_\theta$. After that, we use the fact that $\sum_{a'} \frac{\partial \pi_\theta(a')}{\partial r_\theta(a)} \lambda_\theta = 0$ to get rid of $\lambda_\theta$ in the derivative $\frac{\partial J_f(\pi_\theta)}{\partial r_\theta(a)}$. In other words, we have

$$
\frac{\partial J_f(\pi_\theta)}{\partial r_\theta(a)} = \sum_{a'} \frac{\partial \pi_\theta(a')}{\partial r_\theta(a)} (r^*(x, a') - r_\theta(x, a')).
$$

Using this gradient result, we can derive a novel bound (Lemma C.6) of the gap $J_f(\pi^*) - J_f(\pi_r)$ with a square on the estimation error on the reward function $r$ (Lemma C.6):

$$
\begin{aligned}
&J_f(\pi^*) - J_f(\pi_r) \\
&\leq \eta\gamma C(f, \mathcal{R}_\Theta, \eta) \sum_{a \in \mathcal{A}} \pi_{r'}(a)(r^*(a) - r(a))^2,
\end{aligned}
$$

where $\gamma \in [0, 1]$ is some constant and $r' = \gamma r + (1 - \gamma)r^*$. When $r(a) \geq r^*(a)$, the regret of policy $\pi_r$ can be further bound by

$$
\begin{aligned}
&J_f(\pi^*) - J_f(\pi_r) \\
&\leq \eta C(f, \mathcal{R}_\Theta, \eta) \mathbb{E}_{x \sim \rho, a \sim \pi_r}[(r^*(x, a) - r(x, a))^2].
\end{aligned}
$$

Combine the above result with the Confidence Bound Lemma D.3, the regret of each step in Algorithm 1 can be bounded by

$$
\begin{aligned}
&J_f(\pi^*) - J_f(\pi_t) \\
&\leq 4\eta\mathcal{C}(f, \mathcal{R}_\Theta, \eta) \mathbb{E}_{a^1 \sim \pi_t, a^2 \sim \pi_{t-1}}[(b_{t-1}(x, a^1, a^2))^2].
\end{aligned}
$$

Aggregating the step-regret leads to $\text{Regret}_f(T) = \mathcal{O}(\eta\mathcal{C}(f, \mathcal{R}_\Theta, \eta) \log(N_\mathcal{R} T/\delta) d(\mathcal{R}_\Theta, \xi, T))$. The complete proof of Theorem 3.3 can be found in Appendix D. ∎

*Remark* 3.6. Our last remark about Theorem 3.3 is that this is based on the preference feedback assumption. In some other cases, one may consider the *absolute reward feedback*, which uses a noisy reward for an action. Our Algorithm 1 and Theorem 3.3 can be easily extended to accommodate the absolute reward feedback, and we defer the discussion to Appendix C.

## 4. Derivative-based Exploration

While the optimism-based algorithm (Algorithm 1) enjoys a favorable $O(\log T)$ regret bound, it requires solving for the uncertainty parameter $U$ in every round to construct the bonus $b_t$. This requirement generally entails nontrivial optimization, which can be computationally demanding and thus restrict its practical applicability.

We propose an alternative algorithm that bypasses explicit uncertainty estimation, and instead exploits the sensitivity of the objective function to implicitly account for data uncertainty and balance exploration and exploitation. Specifically, the optimal solution $\pi_\theta(a|x) = \pi_0(a|x)h(\eta(r_\theta(x, a) - \lambda_\theta(x)))$ indicates that the performance of $\pi_\theta$ is determined by two factors:

(I) the accuracy of the estimated reward $r_\theta(x, a)$, and

(II) the estimation error of $h(\cdot)$ at $\eta(r_\theta(x, a) - \lambda_{r_\theta}(x))$ caused by the estimation error of $r_\theta(x, a)$.

Importantly, the second factor depends not only on $r_\theta(x, a)$ but also on the specific function $f$. To see this, we start with

$$
\begin{aligned}
\pi_\theta(a|x) - \pi_{\theta'}(a|x) &\approx \pi_0(a|x)h'(\eta(r_\theta(x, a) - \lambda_\theta(x))) \\
&\quad \cdot \eta(r_{\theta'} - r_\theta + \lambda_\theta - \lambda_{\theta'}).
\end{aligned}
$$

We can see that under the same estimation error of $r_\theta(x, a)$, a larger value of $h'(r_\theta(x, a) - \lambda(x))$ will lead to more uncertainty of $\pi_\theta(a|x)$. On the other hand, if $h'(r_\theta(x, a) - \lambda(x))$ is very close to zero, the error of $\pi_\theta$ caused by the estimation error of $r_\theta(x, a)$ will be relatively negligible. This illustrates the impact of different functions $f$ on the overall estimation error. To the best of our knowledge, prior works (Zhao et al., 2025a; Xiong et al., 2024a) only consider (I) but not (II).

### 4.1. Sampling Scheme

Based on the above intuition, we define a policy $\pi'_\theta$ to assist in sampling the answer:

$$
\pi'_\theta(a|x) = \frac{1}{\overline{T}_\theta(x)} \pi_0(a|x)h'(\eta(r_\theta(x, a) - \lambda_\theta(x))), \quad (9)
$$

where $\overline{T}_\theta(x) = \sum_a \pi_0(a|x)h'(\eta(r_\theta(x, a) - \lambda_\theta(x)))$. This policy $\pi'_\theta$ only concerns the stability of the function $h(\cdot)$ at point $\eta(r_\theta(x, a) - \lambda_\theta(x))$, and can measure the current uncertainty of each action. It focuses only on exploration, and will be used in Algorithm 2 to sample the answers.

Based on $\pi'_\theta$, we introduce two variant policies $\pi_\theta^+$ and $\pi_\theta^-$:

$$
\pi_\theta^+(a|x) = \frac{1}{Z_\theta^+(x)} \pi'_\theta(a|x) \exp(r_\theta(x, a)),
$$

$$
\pi_\theta^-(a|x) = \frac{1}{Z_\theta^-(x)} \pi'_\theta(a|x) \exp(-r_\theta(x, a)),
$$

where

$$Z_\theta^+(x) = \sum_a \pi_\theta'(a|x) \exp(r_\theta(x,a)),$$

$$Z_\theta^-(x) = \sum_a \pi_\theta'(a|x) \exp(-r_\theta(x,a)).$$

These two variants $\pi_\theta^+$ and $\pi_\theta^-$ help us explore when the estimation of $r_\theta(x,a)$ is highly inaccurate. In particular, if the initial estimation of $r_\theta(x,a)$ deviates significantly from its true value, it may lead to $h'(\eta(r_\theta(x,a) - \lambda_\theta(x)))$ being very small (close to 0), which would greatly discourage exploration. Therefore, directly using $\pi_\theta'$ for exploration might be inefficient in such cases. To address this challenge, we introduce two complementary policies $\pi_\theta^+, \pi_\theta^-$ constructed by factors of $\exp(r_\theta(x,a))$ and $\exp(-r_\theta(x,a))$, respectively. The intuition is that, when $h'(\eta(r_\theta(x,a) - \lambda_\theta(x)))$ is close to zero, we should resort to policies that directly use the reward $r_\theta(x,a)$ to promote exploration in both positive (to cover the case reward $r_\theta(x,a)$ is overestimated) and negative (to cover the case reward $r_\theta(x,a)$ is underestimated) directions.

For any prompt $x \in \mathcal{X}$, our sampling procedure will be a mixture of $\pi_\theta', \pi_\theta^+$ and $\pi_\theta^-$. In each round, with probability $1 - p(x)$, we use $\pi_\theta'$ to sample two action $a^1, a^2$, and with probability $p(x)$, we sample $a^1 \sim \pi_\theta^+, a^2 \sim \pi_\theta^-$, where

$$p(x) = Z_\theta^+(x)Z_\theta^-(x)/\{1 + Z_\theta^+(x)Z_\theta^-(x)\}$$

is a parameter that measures the difference between $\pi_\theta'$ and $\{\pi_\theta^+, \pi_\theta^-\}$.

### 4.2. Algorithm Design

After observing the preference label, we adopt a slightly modified loss function $\mathcal{L}(\theta)$ to perform MLE on the current accessible data:

$$\mathcal{L}(\theta) := -\frac{1}{t} \sum_{i=1}^{t} \omega(x_i) \log \sigma(r_\theta(x_i, a_i^\omega) - r_\theta(x_i, a_i^l)), \tag{10}$$

where $\omega(x) := \{\overline{T}_\theta(x) + Z_\theta^+(x)Z_\theta^-(x)\overline{T}_\theta(x)\}/\overline{Z}_\theta$ and $\overline{Z}_\theta = \mathbb{E}_{x \sim \rho}[\overline{T}_\theta(x) + Z_\theta^+(x)Z_\theta^-(x)\overline{T}_\theta(x)]$. The algorithm can be compactly described in Algorithm 2. Note that in Step 7, the algorithm updates its sampling policies $(\pi_\theta', \pi_\theta^+, \pi_\theta^-)$ based on the estimated $\theta_t$ by calling the optimal policy (Proposition 2.3) with respect to $r_\theta$.

*Remark* 4.1. We note that Algorithm 2 does not involve solving an uncertainty parameter, which is usually required in optimism-based algorithms (e.g., Line 7 of Algorithm 1). Solving such an uncertainty parameter can be computationally expensive and might limit the practical applicability, and Algorithm 2 is designed to bypass this step.

**Algorithm 2** Derivative-based Exploration under *f*-divergence Regularized RLHF

---

1: **Input:** parameter $\eta$, reference policy $\pi_0$, reward function class $\mathcal{R}_\Theta = \{r_\theta | \theta \in \Theta\}$
2: **for** step $t = 1, \ldots, T$ **do**
3:   Sample context $x_t \sim \rho$.
4:   With probability $1 - p(x)$, sample $a_t^1, a_t^2 \sim \pi_\theta'(\cdot|x_t)$; with probability $p(x)$, sample $a_t^1 \sim \pi_\theta^+(\cdot|x_t), a_t^2 \sim \pi_\theta^-(\cdot|x_t)$
5:   Observe preference label $y_i \in \{0, 1\}$ from the preference oracle $\mathcal{O}$.
6:   Compute $\theta_t = \arg\min_{\theta \in \Theta} \mathcal{L}(\theta)$ with $\{(x_i, a_i^1, a_i^2, y_i)\}_{i=1}^t$ and Equation (10)
7:   Update $\pi_\theta(a|x) = \pi_0(a|x)h(\eta(r_{\theta_t}(x,a) - \lambda_\theta(x)))$ according to Proposition 2.3.
8: **end for**

---

*Remark* 4.2. For applications in LLMs, one challenge in implementing Algorithm 2 is to sample answers from $\pi_\theta^+, \pi_\theta^-$. Direct sampling can be computationally expensive when faced with long responses, due to the enormous action space. To address this challenge, the classical autoregressive decoding process in LLMs can be leveraged, which breaks the long sentence into a sequence of token-level decisions, e.g., $\pi_\theta^+(\vec{y}|x) = \pi^+(y_1|x)\pi^+(y_2|y_1, x) \cdots \pi^+(y_K|y_{1:K-1}, x)$ for sentence $\vec{y} = y_1 y_2 \cdots y_K$. Then, the token-wise prediction can be done similarly as Proposition 1 in Liu et al. (2024) which, though focused on KL divergence, can be extended to general *f*-divergence by Proposition 2.3.

### 4.3. Theoretical Analysis

With the carefully designed sampling strategy in Algorithm 2, we derive the following theoretical guarantee for the value function.

**Theorem 4.3.** *The value function $J_f(\pi_\theta)$ defined in Equation* (4) *for Algorithm* 2 *satisfies* $\nabla_\theta J_f(\pi_f^*) = 0$ *and* $\nabla_\theta^2 J_f(\pi_f^*) = -\eta\Sigma_*^1$, *where* $\Sigma_*^1 = \mathbb{E}_{x \sim \rho}\left[\overline{T}_{\theta^*}(x)\text{Cov}_{a \sim \pi_\theta'(\cdot|x)}[\nabla_\theta r^*(x,a)|x]\right]$.

In Theorem 4.3, we establish that the covariance of the statistical error in $\theta$ under the sampling scheme (i.e., $\Sigma_*^1$) is proportional to the inverse Hessian of the value function, i.e., $-\nabla_\theta^2 J_f(\pi^*)$. This alignment favors convergence efficiency along directions where the Hessian has large eigenvalues.

**Proof sketch:** Our proof is based on a new and elegant result that the value function $J_f(\pi_\theta)$'s Hessian matrix at point $\theta = \theta^*$ is equal to $-\eta\Sigma_*^1$ where $\Sigma_*^1 = \mathbb{E}_{x \sim \rho}[\overline{T}_\theta(x)\text{Cov}_{a \sim \pi_\theta'}[\nabla_\theta r_\theta(a)]|_{\theta = \theta^*}]$ is the expectation of the covariance matrix under the distribution of the exploration policy $\pi_{\theta^*}'$. This matrix is dominated by the Gaussian matrix $\Omega$ in Theorem E.1, which is the convergence of the

parameter $\theta$. The complete proof of Theorem 4.3 are stated in Appendix E. ∎

To bound the performance of Algorithm 2, we present the following proposition which can be derived from Theorem 4.3.

**Proposition 4.4.** *Under Assumption 3.1 and the conditions in Proposition 2.3, the value function $J_f(\pi_\theta)$ for Algorithm 2 satisfies*

$$\limsup_{n\to\infty} \mathbb{P}\left(\texttt{SubOpt}(\hat{\pi}) \geq C_0 \frac{d(1+\varepsilon)}{T}\right)$$
$$\leq \mathbb{P}\left(\chi_d^2 \geq d(1+\varepsilon)\right)$$
$$\leq \exp\left(-\frac{d}{2}\left(\varepsilon - \log(1+\varepsilon)\right)\right),$$

*where $C_0 = 2\eta C \mathcal{M}(f, \mathcal{R}_\Theta, \eta) \|\overline{T}_\theta + \overline{T}_\theta Z_\theta^+ Z_\theta^-\|_\infty$ and*

$$\mathcal{M}(f, \mathcal{R}_\Theta, \eta)$$
$$:= \max_{r\in\overline{\mathcal{R}_\Theta}} \max_x \sum_{a\in\mathcal{A}} \pi_0(a|x) h'(\eta(r(x,a)-\lambda_r(x))).$$

Note that $\mathcal{M}(f, \mathcal{R}_\Theta, \eta)$ is a constant similar to $C(f, \mathcal{R}_\Theta, \eta)$ (in Theorem 3.3), which also measures how strongly the optimal policy changes when the reward estimate is perturbed. The formal definition is given in Definition E.2 and we will further discussed in Section 5.

This proposition states that when $T$ is sufficiently large, with at least probability $\exp\left(-\frac{d}{2}\left(\varepsilon - \log(1+\varepsilon)\right)\right)$, we have that $\texttt{SubOpt}_f(\hat{\pi}) \leq C_0 \frac{d(1+\varepsilon)}{T}$ for some constant $C_0$. Proposition 4.4 guarantees a sample complexity of $O(1/T)$ for Algorithm 2.

**Proof sketch:** With the result in Theorem 4.3, we can eliminate $\Sigma_*^1$ in the Taylor expansion of $J_f(\pi_\theta)$ at point $\theta = \theta^*$ and get a estimation that $n(J_f(\pi^*) - J_f(\hat{\pi})) \xrightarrow{d} X$, and

$$X \leq 2\eta C \mathcal{M}(f, \mathcal{R}_\Theta, \eta) \|\overline{T}_\theta + \overline{T}_\theta Z_\theta^+ Z_\theta^-\|_\infty \cdot z^\top z,$$

where $z^\top z$ will follow a chi-square distribution with $d$ degree freedom. Then we have the desired result. The complete proof of Proposition 4.4 is given in Appendix E. ∎

## 5. Generic vs. Specific Function $f$

In our theoretical analysis, the influence of the choice of $f$ appears through two factors: $C(f, \mathcal{R}_\Theta, \eta)$ in Theorem 3.3, and $\mathcal{M}(f, \mathcal{R}_\Theta, \eta)$ in Proposition 4.4, which determine the pre-log coefficient of our bounds. The explicit values of $C(f, \mathcal{R}_\Theta, \eta)$ and $\mathcal{M}(f, \mathcal{R}_\Theta, \eta)$ depend on the chosen $f$ as well as the involved problem parameters, and it is generally difficult to explicit calculate them. To shed some light, we discuss some specific examples in the following:

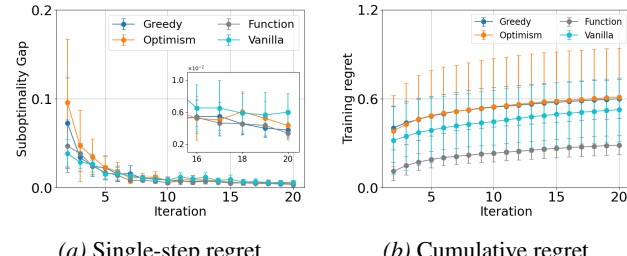

*(a)* Single-step regret      *(b)* Cumulative regret

*Figure 1.* The performance of the two proposed algorithms under chi-squared mixed KL regularization.

● With $f = x\log x$ (i.e., KL divergence), it holds that $C(f, \mathcal{R}_\Theta, \eta) = \mathcal{M}(f, \mathcal{R}_\Theta, \eta) = 1$;

● With $f = x\log x + (x-1)^2$ (i.e., chi-squared mixed KL) or $f = x\log x - \log x$, it holds that $\mathcal{M}(f, \mathcal{R}_\Theta, \eta) \leq C(f, \mathcal{R}_\Theta, \eta) < 1$;

● With $f = -\log x$ (i.e., forward-KL), it holds that $C(f, \mathcal{R}_\Theta, \eta) \geq \mathcal{M}(f, \mathcal{R}_\Theta, \eta) \geq 1$.

The above results for $C, \mathcal{M}$ are proven in Appendix F. Based on these results, it can be observed that with KL divergence, the previous results for KL-regularized RLHF in Zhao et al. (2025a) and Feng et al. (2025) are recovered with Theorem 3.3 and Proposition 4.4, respectively. Also, with $f = x\log x + (x-1)^2$ and $f = x\log x - \log x$, both $C(f, \mathcal{R}_\Theta, \eta)$ and $\mathcal{M}(f, \mathcal{R}_\Theta, \eta)$ are smaller than the corresponding values under KL divergence (i.e., 1). Thus, the obtained upper bounds for regret and sub-optimality under these divergence choices are tighter than those for KL divergence. This important conclusion is also empirically validated by the performance comparisons in Section 6. On the contrary, the values of $C(f, \mathcal{R}_\Theta, \eta)$ and $\mathcal{M}(f, \mathcal{R}_\Theta, \eta)$ are larger than 1 for forward-KL, indicating looser bounds compared with KL divergence.

## 6. Experimental Results

Experiments are conducted to corroborate the theoretical findings, especially the efficiency of the proposed optimism- and derivative-based exploration algorithms. We consider linear scenarios with randomly sampled context vectors and 10 fixed actions for the purpose of demonstration. In particular, we set the BT model with dimension 25. Two choices of $f$-divergence are considered: the first one is the chi-squared mixed KL in Table 1, i.e., $f = (x-1)^2 + x\log x$, and the second one is $f = x\log x - \log x$. Detailed experimental setups and implementation details are deferred to Appendix G.

### 6.1. Baseline

We particularly focus on the comparison among the two proposed algorithms, greedy sampling, and vanilla uni-

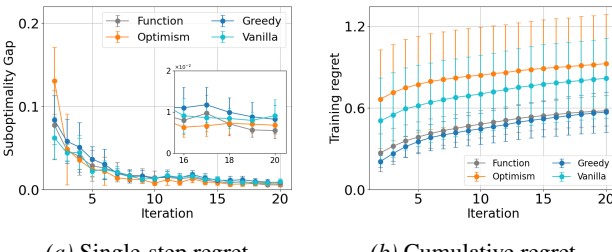

*(a)* Single-step regret      *(b)* Cumulative regret

**Figure 2.** The performance of the two proposed algorithms under $f = x \log x - \log x$ regularization.

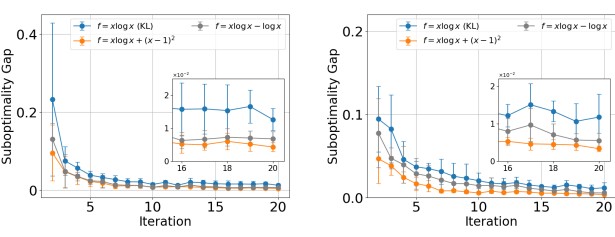

*(a)* Optimism-based design      *(b)* Derivative-based design

**Figure 3.** Performance comparisons of different $f$'s under two proposed algorithms.

form sampling, under different $f$-divergence regularization. Here, greedy sampling means that we directly use the estimated reward to update the sampling policy. To implement optimism (Algorithm 1), we consider the BT model with linear rewards, as considered in Xiong et al. (2023); Zhao et al. (2025a) where closed-form optimism is adopted. In particular, at each step $t$, the optimistic estimate $\hat{r}_t(x, a) + \beta ||\phi(x, a) - \mathbb{E}_{x \sim d_0}[\phi(x, \pi_0)]||_{\Sigma_t^{-1}}$ is leveraged to construct $\pi^1 = \pi_t$ for sampling the first answer while the second answer is obtained from $\pi^2 = \pi_{t-1}$ as in Algorithm 1, where $\phi(x, a)$ is the linear feature of the context-action pair and $\Sigma_t$ is the regularized Gram matrix from the collected data. Specifically, we choose $\beta = 0.1$ as the level of optimism. Note that when $\beta = 0$, the bonus terms vanish, which returns to greedy sampling. For the second proposed algorithm (Algorithm 2), we implement exactly as described in Section 4 and in Figure 1 we mark it as "Function".

### 6.2. Results

The experimental results are plotted in Figures 1 and 2. We can see that both of our algorithms have a good convergence performance, which demonstrates their efficiency. Furthermore, Algorithm 2 that uses the derivative for exploration achieves a better performance. This interesting observation suggests that "derivative as uncertainty" not only has intriguing theoretical properties as discussed in Section 4.3, but also possesses practical utility that may be worth further investigation. Additionally, greedy sampling performs competitively due to the fact that sampling directly with

respect to the estimated reward is stochastic as shown in Proposition 2.3, and thus has some intrinsic exploration ability. This observation has been similarly reported in Wu et al. (2025) for the case of KL divergence. Furthermore, Figure 3 compares the performance of the two proposed algorithms under different choices of $f$. The results show that using the chi-squared–mixed KL divergence (i.e., $f(x) = x \log x + (x-1)^2$) and $f(x) = x \log x - \log x$ leads to smaller suboptimality gaps than using the standard KL divergence. This observation is consistent with the theoretical discussion in Section 5, which predicts smaller constants $\mathcal{C}(f, \mathcal{R}_\Theta, \eta)$ and $\mathcal{M}(f, \mathcal{R}_\Theta, \eta)$ for these two choices.

## 7. Related Works

**Theoretical studies on online RLHF.** In recent years, a relatively comprehensive theoretical understanding of canonical bandits and reinforcement learning (RL) has been developed (Lattimore & Szepesvári, 2020; Agarwal et al., 2019). These works laid the foundation for regret analysis and policy optimization in sequential decision making. To achieve sublinear regret in the online RLHF, one of the most widely adopted approaches is to use optimistic estimates in the face of uncertainty (Xiong et al., 2023; Zhao et al., 2025a). For instance, Zhao et al. (2025a) establishes $O(\log(T))$ regret guarantees by constructing confidence bounds based on an uncertainty bonus. Xiong et al. (2023) considers uncertainty-driven exploration without explicit bonuses: choosing policies maximizing uncertainty relative to past data, which effectively balances exploration and exploitation more implicitly. Despite their favorable performance, those algorithms cost a lot of computation in handling the uncertainty, such as solving complex optimization problems. Recent work began to seek more computationally efficient ways while still guaranteeing good performance. Feng et al. (2025) proposed a mixed sampling schema that balances exploration without explicitly handling the uncertainty, but this approach is still limited in the KL-regularized setting. Existing works commonly only consider KL divergence regularization or do not consider regularization, and our work will give unified analyses to general $f$-divergence regularization.

***f*-divergence in RLHF.** Beyond reverse KL divergence, some recent works have started to explore leveraging alternative choices of divergence as regularization in RLHF. Shan et al. (2025) argued that forward KL regularization can provide favorable optimization properties in preference-based learning. Using chi-squared mixed KL divergence has been proved efficient and robust to the over-optimization phenomenon in Huang et al. (2025). Also, $\alpha$-divergence (Belousov & Peters, 2018) has been studied as a one-parameter family of $f$-divergence that can provide tradeoffs between the training process. With more and more kinds of diver-

gence being demonstrated beneficial, it becomes natural that construct a unified theory for general $f$-divergence regularized training. These works motivate the need for a unified theoretical treatment of general $f$-divergences in RLHF, which, unfortunately, is currently lacking. Wang et al. (2024); Sun et al. (2025) combines $f$-divergence with Direct Preference Optimization (DPO) (Rafailov et al., 2023) and demonstrated that alternative divergences can lead to training diversity, but without theoretical validation. Aminian et al. (2026) establish an $O(1/\sqrt{T})$ sample complexity for forward KL and an $O(1/T)$ sample complexity for reverse KL, whereas this work achieves $O(1/T)$ rates for both cases as well as for additional divergence choices through a unified analysis. They also consider a divergence-free notion of performance gap, which captures a different notion of optimality and is therefore beyond the scope of the present analysis. Zhao et al. (2025b) get an $O(\epsilon^{-1})$ sample complexity bound in $f$-divergence regularized RLHF but still limit in the *offline RLHF* setting. To the best of our knowledge, our work is the first result providing provable sampling schemas for online $f$-divergence regularized RLHF.

## 8. Conclusion

This work investigated RLHF under a general $f$-divergence regularized contextual bandits framework. We have proposed two effective sampling algorithms for general $f$-divergence regularized RLHF. The first algorithm leverages the uncertainty of collected data to construct a bonus to realize optimism, while the second one interprets the derivative of the function as the uncertainty to build the exploration policy, which is the first time such an interpretation is introduced for uncertainty estimation. Theoretical analysis revealed that both algorithms achieve $O(1/T)$ suboptimal gap and $O(\log(T))$ regret, representing the first time that such favorable orders are proven for general $f$-divergence regularized RLHF in an online setting. Simulation results further corroborated the effectiveness of the two algorithms.

## Acknowledgements

This work is supported in part by the National Science Foundation (NSF) under grants CPS-2313110, ECCS-2143559, and ECCS-2531023, and the University of Virginia Grand Challenge Research Investments – Digital Technology Smart Infrastructure (Strategic Investment Fund Award #200).

## Impact Statement

This work focuses on the theoretical study of reinforcement learning from human feedback (RLHF) and proposes two efficient methods to train the language model under a general $f$-divergence regularized setting. While acknowledging the need for responsible usage of the proposed methods, we do not foresee major negative societal impacts due to the theoretical nature of this work.

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

## A. Limitations and Future Directions

The following are a few directions worth further investigation, beyond the scope of the current paper.

- From the theoretical perspective, it would be interesting to further tighten the obtained performance bounds of the two sampling algorithms. In particular, as illustrated in the proofs, the bounds contain multiplicative constants of $\mathcal{C}(f, \mathcal{R}_\Theta, \eta)$ and $\mathcal{M}(f, \mathcal{R}_\Theta, \eta)$, which may be worth further studies on its necessity.

- The application of $f$-divergence regularized RLHF can be further investigated. By varying the choice of the divergence function $f$, one can induce different structural biases in the resulting target policy, which in turn influences both its learning dynamics and final performance. Each $f$ encodes a distinct trade-off between exploration, conservatism, and robustness, thereby shaping how the policy adapts to human feedback.

- This work focuses on the single-step setting, i.e., a contextual bandit problem, as it is the most widely adopted RLHF scenario. Still, with a growing amount of interest on performing post-training in multi-turn scenarios (Xiong et al., 2024b), it would be a promising direction to extend the theoretical results in this work to multi-step RL.

## B. Properties of $f$-divergence

### B.1. Basic Properties

Recall the definition of $f$-divergence is

$$D_f(p||q) = \mathbb{E}_{q(x)}\left[f\left(\frac{p(x)}{q(x)}\right)\right],$$

where we require $f$ is convex and $f(1) = 0$. Due to the convexity of $f$, the $f$-divergence is always non-negative.

**Lemma B.1.** *For any convex $f$, we have*

$$D_f(p||q) \geq 0,$$

*Moreover, if $f$ is strictly convex at $1$, then $D_f(p||q) = 0$ only when $p = q$.*

*Proof.* Since $f$ is convex, Jensen's inequality suggests

$$D_f(p||q) = \mathbb{E}_{q(x)}\left[f\left(\frac{p(x)}{q(x)}\right)\right] \geq f\left(\mathbb{E}_{q(x)}\left[\frac{p(x)}{q(x)}\right]\right) = f(1) = 0,$$

which concludes the proof. $\square$

Furthermore, the $f$-divergence is jointly convex over $p$ and $q$.

**Lemma B.2.** *Given any $\alpha \in [0, 1]$, we have*

$$D_f(\alpha p_1 + (1-\alpha)p_2 || \alpha q_1 + (1-\alpha)q_2) \leq \alpha D_f(p_1||q_1) + (1-\alpha)D_f(p_2||q_2).$$

*Proof.* Refer to the proof of Proposition B.3. in Zhang (2023). $\square$

### B.2. Optimal Policy under $f$-divergence Regularization

In $f$-divergence regularization, for the reward function $r$, we want to solve the following optimization problem:

$$\pi_r = \arg\max_\pi \mathbb{E}_{x\sim\rho}\mathbb{E}_{a\sim\pi(\cdot|x)}r(x,a) - \frac{1}{\eta}D_f(\pi, \pi_0|x). \tag{11}$$

In Proposition 2.3, we obtain that

$$\pi_r(a|x) = \pi_0(a|x)f'^{-1}\left(\eta(r(x,a) - \lambda_r)\right),$$

and we give the proof of Proposition 2.3 below.

*Proof of Proposition 2.3.* We use the Karush-Kuhn-Tucker (KKT) conditions to solve the given optimization problem. We fix the context $x$, and the Lagrangian function can be constructed as follows:

$$\mathcal{L}(\pi, \lambda, \alpha) = \sum_a \pi(a|x) r(x, a) - \frac{1}{\eta} \sum_a \pi_0(a|x) f\left(\frac{\pi(a|x)}{\pi_0(a|x)}\right) - \lambda \left(\sum_a \pi(a|x) - 1\right) + \alpha(a)\pi(a|x).$$

The KKT condition suggests that

$$\begin{cases} r(x, a) - \frac{1}{\eta}\frac{\partial}{\partial \pi(a|x)}\mathbb{E}_{\pi_0}\left[f\left(\frac{\pi(a|x)}{\pi_0(a|x)}\right)\right] - \lambda + \alpha(y) = 0, \ \forall a \in \mathcal{A} \\ \alpha(a) \geq 0, \ \forall a \in \mathcal{A} \\ \alpha(a) \cdot \pi(a|x) = 0, \ \forall a \in \mathcal{A} \end{cases}.$$

We have, for each $a \in \mathcal{A}$,

$$r(x, a) - \frac{1}{\eta} f'\left(\frac{\pi_r(a|x)}{\pi_0(a|x)}\right) - \lambda + \alpha(a) = 0.$$

We can solve this for $\pi(a|x)$, assuming that the inverse of $f'$ exists:

$$\pi_r(a|x) = \pi_0(a|x) f'^{-1}\left(\eta(r(x, a) - \lambda + \alpha(a))\right).$$

Assume that $\pi_0(a|x) > 0$ for any valid $x$ and $f'$ is invertible with $\text{dom}(f') \in \mathbb{R}_+$, the solution $\pi_r(a|x) > 0$. That means $\alpha(a)$ must be zero. For each $x$, the solution can be expressed as

$$\pi_r(a|x) = \pi_0(a|x) f'^{-1}\left(\eta(r(x, a) - \lambda_r(x))\right),$$

and $\lambda_r(x)$ is determined by $\sum_a \pi_{r_\theta}(a|x) = 1$, that is,

$$\sum_a \pi_0(a|x) f'^{-1}\left(\eta(r(x, a) - \lambda_r(x))\right) = 1.$$

Furthermore, the existence of $f'$ suggests that $f'^{-1}$ is strictly increasing, and the solution of $\lambda$ is unique. This implies the solution to the KKT condition is unique and will be the solution to Equation (11).

Therefore, for those function $f$ that $0 \notin \text{dom}(f')$ with the assumption that $\pi_0(a|x) > 0$ almost surely, we have $\pi_r(a|x) = \pi_0(a|x) f'^{-1}\left(\eta(r(x, a) - \lambda_r(x))\right), \ \forall a$. In particular, the reverse KL, forward KL, and the JS divergences are among this category. We refer to Table 1 for details. $\qquad \square$

In the BT model, any two reward functions $r_1$, $r_2$ that differ only by a constant, i.e., $r_1(x, a) - r_2(x, a) = A(x), \forall x, a$, induce the same probability model:

$$P_1(a^1 \succeq a^2|x) = \sigma(r_1(x, a^1) - r_1(x, a^2)) = \sigma(r_2(x, a^1) - r_2(x, a^2)) = P_2(a^1 \succeq a^2|x).$$

We have the following theorem that shows, for the optimal policy under $f$-divergence regularization, any two reward functions $r_1$, $r_2$ that differ only by a constant will also induce the same optimal policies $\pi_{r_1} = \pi_{r_2}$.

**Theorem B.3.** *If $\pi_0(a|x) > 0$ for any valid $x$ and $f'$ is invertible with $\text{dom}(f') \in \mathbb{R}_+$, any two reward functions $r_1$, $r_2$ that differ only by a constant, i.e., $r_1(x, a) - r_2(x, a) = A(x)$, induce the same optimal policy.*

*Proof.* Since $f$ is convex and $f'$ is invertible, $f'^{-1}$ will be strictly increasing (because it is the inverse function of $f'$). For each $x$, we can get that $\lambda(x)$ is the solution of

$$F(\lambda) = \sum_a \pi_0(a|x) f'^{-1}(\eta(r(x, a) - \lambda)) = 1,$$

which is unique. This means any two reward functions $r_1$, $r_2$ that differ only by a constant will also induce the same optimal policies. $\qquad \square$

## C. Optimism-based Exploration with Reward Feedback

### C.1. Algorithm Design

In this section, we give the respective results in Section 3 in the absolute feedback case. In the absolute feedback setting, we will return noisy reward feedback during the interaction with the environment, instead of sending preference data.

For Algorithm 3, in each iteration $t$, we sample a context $x_t$ from $\rho$ and sample an action from policy $\pi_t(\cdot|x_t)$ (Step 4). After getting a noisy reward feedback $\tilde{r}_t$, we use maximum likelihood estimation on previous data to get an estimated $\theta_t$ (step 6). Finally, an updated policy $\pi_{t+1}$ is obtained as the optimal policy corresponding to the optimistic reward estimate $\hat{r}_t$.

---

**Algorithm 3** Optimism-based Exploration with Reward Feedback

---

1: **Input:** parameter $\eta$, reference policy $\pi_0$, random noise $\epsilon$, reward function class $\mathcal{R}_\Theta$
2: Initialize $\pi_t \leftarrow \pi_0$
3: **for** $t = 1, \ldots, T$ **do**
4:     Sample context $x_t \sim \rho$ and two actions $a_t \sim \pi_t(\cdot|x_t)$
5:     Observe reward $\tilde{r}_t = r^*(x_t, a_t) + \epsilon_t$, where $\epsilon_t$ is the random noise.
6:     Perform the maximum likelihood estimation with $\{(x_i, a_i, r_i)\}_{i=1}^t$:

$$\theta_t \leftarrow \operatorname*{arg\,min}_{\theta \in \Theta} \sum_{i \in [t]} (r_\theta(x_i, a_i) - \tilde{r}_i)^2.$$

7:     Use the optimistic reward model $\hat{r}_t(x, a) := r_{\theta_t}(x, a) + b_t^{\mathrm{RF}}(x, a)$ to obtain the optimal policy as

$$\pi_{t+1}(\cdot|\cdot) = \pi_0(\cdot|\cdot)h(\eta(\hat{r}_t(\cdot|\cdot) + \lambda(\cdot))).$$

8: **end for**

---

### C.2. Theoretical Analysis

In the case of absolute reward feedback, we establish a novel regret bound for Algorithm 3 for online RLHF with $f$-divergence regularization. The standard realizability assumption is the same as Assumption 3.1, which indicates that the true reward model $r^*$ is in the considered reward class $\mathcal{R}_\Theta$.

Before presenting the main theorem of this section, we give the definition of Eluder dimension under the absolute reward feedback setting, which is commonly adopted in Zhao et al. (2025a) for measuring the effect of out-of-sample data.

**Definition C.1** (Eluder Dimension, Bradley-Terry Model). Under the Bradley-Terry model, for any sequence $\mathcal{D}_{t-1} = \{(x_i, a_i, \hat{r}_i)\}_{i=1}^{t-1}$, we define the uncertainty of $(x, a)$ with respect to $\mathcal{R}_\Theta = \{r_\theta : \theta \in \Theta\}$ as:

$$U_{\mathrm{RF}}(\xi, x, a; \mathcal{R}_\Theta, \mathcal{D}_{t-1}) = \sup_{r_{\theta_1}, r_{\theta_2} \in \mathcal{R}_\Theta} \frac{|r_{\theta_1}(x, a) - r_{\theta_1}(x, a)|}{\sqrt{\xi + \sum_{i=1}^{t-1}(r_{\theta_1}(x, a_i) - r_{\theta_1}(x, a_i))^2}},$$

and the corresponding Eluder dimension as

$$d_{\mathrm{RF}}(\mathcal{R}_\Theta, \xi, T) := \sup_{x_{1:T}, a_{1:T}} \sum_{t \in [T]} \min\{1, [U_{\mathrm{RF}}(\xi, x, a_t; \mathcal{R}_\Theta, \mathcal{D}_{t-1})]^2\}.$$

To construct the confidence bound, we define the following bonus:

$$b_t^{\mathrm{RF}}(x, a) = \min\{1, \beta_T^{\mathrm{RF}} \cdot U(\xi, x, a; \mathcal{R}_t^{\mathrm{RF}}, \mathcal{D}_t)\},$$

where $\beta_T^{\mathrm{RF}} = 16\log(N_\mathcal{R} T/\delta)$, and the confidence set $\mathcal{R}_t$ is

$$\mathcal{R}_t^{\mathrm{RF}} = \{r_\theta \in \mathcal{R}_\Theta : (r_\theta(x, a) - r_{\theta_t}(x, a))^2 + \xi \le (\beta_T^{\mathrm{RF}})^2\}.$$

With the above definitions, the following lemma shows that $b_t^{\mathrm{RF}}(x, a)$ is a valid bonus.

**Lemma C.2.** *Under Algorithm 3, we have with probability at least $1 - \delta$ for all $t \in [T]$, the uniform optimism event that*

$$\mathcal{E}_t^{\mathrm{RF}} = \{r_{\theta_t}(x, a) + b_t^{\mathrm{RF}}(x, a) - r^*(x, a) > 0, \ \forall(x, a) \in \mathcal{X} \times \mathcal{A}\}$$

*holds true.*

The proof of Lemma C.2 can be found in Lemma A.3 in Zhao et al. (2025a).

Now we present the main result in this section.

**Theorem C.3.** *Under Assumption 3.1, for any $\delta > 0$, by taking $\beta_T = 16 \log(N_{\mathcal{R}} T/\delta)$, with probability at least $1 - \delta$, the output of Algorithm 3 satisfies*

$$\text{Regret}_f^{\text{RF}}(T) = \mathcal{O}(\eta \mathcal{C}(f, \mathcal{R}_\Theta, \eta) \log(N_{\mathcal{R}} T/\delta) d_{\text{RF}}(\mathcal{R}_\Theta, \xi, T)).$$

*Proof.* Conditioning on the event $\cup_{t \in [T]} \mathcal{E}_t^{\text{RF}}$ in Lemma C.2, we have $r_{\theta_t} + b_t^{\text{RF}} \geq r^*$. Using Lemma C.6, we can directly get our result as follows.

$$\begin{aligned}
\text{Regret}_f^{\text{RF}}(T) &= \sum_{t=1}^T J_f(\pi^*) - J_f(\pi_t) \\
&\leq 4 \sum_{t=1}^T \eta \mathcal{C}(f, \mathcal{R}_\Theta, \eta) \mathbb{E}_{x \sim \rho, \, a \sim \pi_t}[(b_{t-1}^{\text{RF}}(x, a))^2] \\
&= 4 \eta \mathcal{C}(f, \mathcal{R}_\Theta, \eta)(\beta_T^{\text{RF}})^2 \sum_{t=1}^T \mathbb{E}_{x \sim \rho, \, a \sim \pi_t} \min\left\{1, (U_{\text{RF}}(\xi, x_t, a_t; \mathcal{R}_{t-1}, \mathcal{D}_{t-1}))^2\right\} \\
&= \mathcal{O}\left(\eta \mathcal{C}(f, \mathcal{R}_\Theta, \eta) \log(N_{\mathcal{R}} T/\delta) d_{\text{RF}}(\mathcal{R}_\Theta, \xi, T)\right),
\end{aligned}$$

where the second equality uses the definition of Eluder dimension (Definition C.1) and concludes the proof. □

Theorem C.3 guarantees a logarithmic regret bound for optimistic sampling in Algorithm 3. Notably, this theorem achieves the same order as the preference feedback version (Theorem 3.3). For any choice of $f$-divergence, the algorithm shares the same regret order as the one achieved by Zhao et al. (2025a) under the reverse KL divergence, which further validates the efficiency of the proposed design.

The following is devoted to the proof of the key Lemma C.6. Before going further, we present some facts about the value function and the definition of the constant $\mathcal{C}(f, \mathcal{R}_\Theta, \eta)$.

**Lemma C.4.** *Any two reward functions $r_1$, $r_2$ that differ only by a constant, i.e., $r_1(x, a) - r_2(x, a) = A(x)$, induce the same objective value $J_f(\pi_{r_1}) = J_f(\pi_{r_2})$.*

*Proof.* Theorem B.3 shows the two reward functions $r_1$ and $r_2$ induce the same optimal policy. By the definition of objective value in Equation (4), we obtain $J_f(\pi_{r_1}) = J_f(\pi_{r_2})$. □

We now introduce a notation that measures the difference between $h$ and $h'$ where $h = (f')^{-1}$, which establishes the bridge between $f'^{-1}$ and $(f'^{-1})'$.

**Definition C.5.** For a given function $f$ and parameter class $\Theta$, we define

$$\mathcal{C}(f, \mathcal{R}_\Theta, \eta) = \max_{r \in \overline{\mathcal{R}_\Theta}} \max_{x, a} \frac{h'(\eta(r(x, a)) - \lambda_r(x))}{h(\eta(r(x, a)) - \lambda_r(x))}.$$

where $\overline{\mathcal{R}_\Theta}$ is the convex hull of $\mathcal{R}_\Theta$.

With the above notations, we present the decomposition lemma for the value function $J_f(\pi)$.

**Lemma C.6** (Value Decomposition Bound). *For any reward functions $r$, the value function $J_f(\pi_r) = \mathbb{E}_{x \sim \rho}[\mathbb{E}_{a \sim \pi_r} r^*(x, a) - D_f(\pi_r, \pi_0 | x)]$ satisfies*

$$J_f(\pi_f^*) - J_f(\pi_r) \leq \eta \gamma C(f, \mathcal{R}_\Theta, \eta) \mathbb{E}_{x \sim \rho, a \sim \pi_{r'}}[(r^*(x, a) - r(x, a))^2],$$

*where $\gamma$ is a constant in $[0, 1]$ and $r' = \gamma r + (1 - \gamma) r^*$. Furthermore, if $r$ dominates the real reward $r^*$, i.e., $r(x, a) \geq r^*(x, a)$ for all $(x, a) \in (\mathcal{X}, \mathcal{A})$, we have*

$$J_f(\pi_f^*) - J_f(\pi_r) \leq \eta C(f, \mathcal{R}_\Theta, \eta) \mathbb{E}_{x \sim \rho, a \sim \pi_r}[(r^*(x, a) - r(x, a))^2].$$

*Proof.* Recall that we have an optimal policy $\pi_r$ for a reward function $r : \mathcal{X} \times \mathcal{A} \to [0,1]$

$$\pi_r(a|x) = \pi_0(a|x)(f')^{-1}(\eta(r(x,a) - \lambda_r(x)))$$

and for each $x$, $\lambda_r(x)$ is determined by $r$ through

$$\sum_a \pi_0(a|x)(f')^{-1}(\eta(r(x,a) - \lambda_r(x))) = 1. \tag{12}$$

Taking the derivative on both sides of Equation (12), for each $x$, we can get the relation of $\lambda_r(x)$ and $r(x,a)$ as

$$\frac{\partial \lambda_r(x)}{\partial r(x,a_0)} = \frac{T_r(x,a_0)}{\sum_a T_r(x,a)},$$

where

$$T_r(x,a) = \pi_0(a|x)((f')^{-1})'(\eta(r(x,a) - \lambda_r(x))),$$

$$\frac{\partial \pi_r(a|x)}{\partial r(x,a_0)} = \eta \pi_0(a|x)((f')^{-1})'(\eta(r(x,a) - \lambda_r(x)))(\mathbf{1}\{a = a_0\} - \frac{\partial \lambda_r(x)}{\partial r(x,a_0)}) = \eta T_r(x,a)\left(\mathbf{1}\{a = a_0\} - \frac{\partial \lambda_r(x)}{\partial r(x,a_0)}\right).$$

Now, for each reward function $r$,

$$J_f(\pi_r) = \mathbb{E}_{x \sim \rho}[\mathbb{E}_{a \sim \pi_r} r^*(x,a) - \eta^{-1}\mathbb{E}_{a \sim \pi_0} f(\frac{\pi_r(a|x)}{\pi_0(a|x)})]$$

$$= \mathbb{E}_{x \sim \rho}\left[\sum_a \pi_r(a|x)r^*(x,a) - \eta^{-1}\sum_a \pi_0(a|x)f(\frac{\pi_r(a|x)}{\pi_0(a|x)})\right].$$

$$\frac{\partial J_f(\pi_r)}{\partial r(x,a_0)} = \mathbb{E}_{x \sim \rho}\sum_a \frac{\partial \pi_r(a|x)}{\partial r(x,a_0)}\left(r^*(x,a) - \eta^{-1}\pi_0(a|x)\frac{1}{\pi_0(a|x)}f'(\frac{\pi_r(a|x)}{\pi_0(a|x)})\right)$$

$$= \mathbb{E}_{x \sim \rho}\sum_a \frac{\partial \pi_r(a|x)}{\partial r(x,a_0)}\left(r^*(x,a) - r(x,a) + \lambda_r(x)\right),$$

and, for each $x$, we have

$$\sum_a \frac{\partial \pi_r(a|x)}{\partial r(x,a_0)}\lambda_r(x) = \eta \lambda_r(x)(T_r(x,a_0) - T_r(x,a_0)) = 0.$$

Thus, we get

$$\frac{\partial J_f(\pi_r)}{\partial r(x,a_0)} = \mathbb{E}_{x \sim \rho}\sum_a \frac{\partial \pi_r(a|x)}{\partial r(x,a_0)}\left(r^*(x,a) - r(x,a) + \lambda_r(x)\right)$$

$$= \eta \mathbb{E}_{x \sim \rho}\left[T_r(x,a_0)(r^*(x,a_0) - r(x,a_0)) - \eta \sum_a T_r(x,a)\frac{\partial \lambda_r(x)}{\partial r(x,a_0)}(r^*(x,a) - r(x,a))\right].$$

By the mean value theorem, for reward function $r$, there exist $\gamma \in [0, 1]$ and $r' = \gamma r + (1 - \gamma)R$ such that

$$J_f(\pi_f^*) - J_f(\pi_r) = \mathbb{E}_{x \sim \rho} \sum_{a_1 \in \mathcal{A}} \frac{\partial J_f(\pi_{r'})}{\partial r(x, a_1)} (r^*(x, a_1) - r(x, a_1))$$

$$= \eta \gamma \mathbb{E}_{x \sim \rho} \sum_{a \in \mathcal{A}} T_{r'}(x, a)(r^*(x, a) - r(x, a))^2$$

$$- \eta \mathbb{E}_{x \sim \rho} \sum_{a_1 \in \mathcal{A}} \sum_{a_2 \in \mathcal{A}} T_{r'}(x, a_2) \frac{\partial \lambda_{r'}(x)}{\partial r(x, a_1)} (r^*(x, a_2) - r(x, a_2))(r^*(x, a_1) - r(x, a_1))$$

$$= \eta \gamma \mathbb{E}_{x \sim \rho} \sum_{a \in \mathcal{A}} T_{r'}(a)(r^*(a) - r(a))^2$$

$$- \eta \gamma \mathbb{E}_{x \sim \rho} \left( \sum_{a \in \mathcal{A}} T_{r'}(x, a)(r^*(x, a) - r(x, a)) \right)^2 / \left( \sum_{a \in \mathcal{A}} T_{r'}(x, a) \right)$$

$$\leq \eta \mathbb{E}_{x \sim \rho} \gamma \sum_{a \in \mathcal{A}} T_{r'}(x, a)(r^*(x, a) - r(x, a))^2$$

$$\leq \eta \mathbb{E}_{x \sim \rho} \gamma C(f, \mathcal{R}_\Theta, \eta) \sum_{a \in \mathcal{A}} \pi_{r'}(a|x)(r^*(x, a) - r(x, a))^2,$$

which concludes the first part of the proof.

To focus on the monotonicity of $\gamma$, we calculate the derivative. Define $U(x; \gamma) = \gamma \sum_{a \in \mathcal{A}} \pi_{r'}(a|x)(r^*(x, a) - r(x, a))^2$.

$$\frac{\partial U(x; \gamma)}{\partial \gamma} = \sum_{a \in \mathcal{A}} \left[ \pi_{r'}(a|x)(r^*(x, a) - r(x, a))^2 + \frac{\partial \pi_{r'}(a|x)}{\partial \gamma} \gamma (r^*(x, a) - r(x, a))^2 \right],$$

and

$$\frac{\partial \pi_{r'}(a|x)}{\partial \gamma} = \sum_{a'} \frac{\partial \pi_{r'}(a|x)}{\partial r'(x, a')} \frac{\partial r'(x, a')}{\partial \gamma}$$

$$= \sum_{a'} \left[ \pi_0(a|x) h'(\eta(r'(x, a) - \lambda_{r'}(x))) \eta (\mathbf{1}\{a' = a\} - \frac{\partial \lambda_{r'}(x)}{\partial r'(x, a')}) \right] (r(x, a') - r^*(x, a'))$$

$$= \eta \left[ \pi_0(a|x) h'(\eta(r'(x, a) - \lambda_{r'}(x))) \right] \left( (r(x, a) - r^*(x, a)) - \mathbb{E}_{a' \sim \pi_{r'}'} [r(x, a') - r^*(x, a')] \right).$$

Thus, we have

$$\frac{\partial U(x; \gamma)}{\partial \gamma} \geq \sum_{a \in \mathcal{A}} \left[ \frac{\partial \pi_{r'}(a|x)}{\partial \gamma} \gamma (r^*(x, a) - r(x, a))^2 \right]$$

$$= \eta \overline{T}_{r'}(x) \left\{ \mathbb{E}_{a \sim \pi_{r'}'} \left[ (r(x, a) - r^*(x, a))^3 \right] \right.$$

$$\left. - \mathbb{E}_{a \sim \pi_{r'}'} \left[ (r(x, a) - r^*(x, a))^2 \right] \mathbb{E}_{a' \sim \pi_{r'}'} [r(x, a') - r^*(x, a')] \right\}$$

$$\geq 0,$$

where the last inequality follows by Lemma H.1. The proof is thus complete. □

## D. Proof of Theorem 3.3

In this section, we present the proof of Theorem 3.3 in Section 3, which establishes a novel regret bound. To prove the theorem, we introduce the following lemmas to bound the error of the maximum likelihood estimates.

**Lemma D.1.** *Given the training data $\mathcal{D} = \{(x_i, a_i^1, a_i^2, y_i)\}_{i=1}^n$, with probability at least $1 - \delta$ and the MLE estimator $r_\theta$ satisfies the following estimation:*

$$\sum_{i=1}^n \left[ r_\theta(x_i, a_i^1) - r_\theta(x_i, a_i^2) - \left( r^*(x_i, a_i^1) - r^*(x_i, a_i^2) \right) \right]^2 \leq 2e \log \frac{N_\mathcal{R}}{\delta}.$$

*Proof.* Denote $P_\theta(x_i, a_i^1, a_i^2) = \sigma(r_\theta(x_i, a_i^1) - r_\theta(x_i, a_i^2))$, we begin with bounding

$$\sum_{i=1}^n (P_\theta(x_i, a_i^1, a_i^2) - P^*(x_i, a_i^1, a_i^2))^2.$$

For any fixed $r \in \mathcal{R}_\Theta$, we first upper bound its logarithmic moment generating function as

$$\log \mathbb{E} \exp \left( \sum_{i=1}^n \log \frac{P(y_i|x_i, a_i^1, a_i^2)}{P^*(y_i|x_i, a_i^1, a_i^2)} \right)$$

$$= \log \mathbb{E} \exp \left( \sum_{i=1}^{n-1} \log \frac{P(y_i|x_i, a_i^1, a_i^2)}{P^*(y_i|x_i, a_i^1, a_i^2)} \right) + \log 2 \mathbb{E}_{y_n|x_n, a_n^1, a_n^2} \sqrt{\frac{P(y_n|x_n, a_n^1, a_n^2)}{P^*(y_n|x_n, a_n^1, a_n^2)}}$$

$$= \log \mathbb{E} \exp \left( \sum_{i=1}^{n-1} \log \frac{P(y_i|x_i, a_i^1, a_i^2)}{P^*(y_i|x_i, a_i^1, a_i^2)} \right) + \log \left( 1 - H\left( P(y_n|x_n, a_n^1, a_n^2) \| P^*(y_n|x_n, a_n^1, a_n^2) \right)^2 \right)$$

$$\leq \log \mathbb{E} \exp \left( \sum_{i=1}^{n-1} \log \frac{P(y_i|x_i, a_i^1, a_i^2)}{P^*(y_i|x_i, a_i^1, a_i^2)} \right) - H\left( P(y_n|x_n, a_n^1, a_n^2) \| P^*(y_n|x_n, a_n^1, a_n^2) \right)^2$$

$$\leq \cdots \leq - \sum_{i=1}^n H\left( P(y_i|x_i, a_i^1, a_i^2) \| P^*(y_i|x_i, a_i^1, a_i^2) \right)^2, \tag{13}$$

where $H(P\|Q)$ is the Hellinger distance defined by

$$H(P\|Q)^2 := \int_\Omega \left( \sqrt{p(z)} - \sqrt{q(z)} \right)^2 d\mu(z).$$

We continue to lower-bound the Hellinger distance by

$$\sum_{i=1}^n \left( H(P(y_i|x_i, a_i^1, a_i^2) \| P^*(y_i|x_i, a_i^1, a_i^2)) \right)^2 \geq \sum_{i=1}^n \left( \mathrm{TV}(P(y_i|x_i, a_i^1, a_i^2) \| P^*(y_i|x_i, a_i^1, a_i^2)) \right)^2$$

$$= \sum_{i=1}^n (P(x_i, a_i^1, a_i^2) - P^*(x_i, a_i^1, a_i^2))^2, \tag{14}$$

where the inequality uses the fact that for any distribution $p, q$, $H(p, q) \geq \mathrm{TV}(p, q)$ according to Theorem B.9 of Zhang (2023).

Then, by invoking Lemma H.3, we obtain for any $R \in \mathcal{R}_\Theta$, with probability at least $1 - \delta$,

$$\sum_{i=1}^n \log \frac{P(y_i|x_i, a_i^1, a_i^2)}{P^*(y_i|x_i, a_i^1, a_i^2)} \leq \log(N_\mathcal{R}/\delta) + \log \mathbb{E} \exp \left( \sum_{i=1}^n \log \frac{P(y_i|x_i, a_i^1, a_i^2)}{P^*(y_i|x_i, a_i^1, a_i^2)} \right)$$

$$\leq - \sum_{i=1}^n H\left( P(y_i|x_i, a_i^1, a_i^2) \| P^*(y_i|x_i, a_i^1, a_i^2) \right)^2 + \log(N_\mathcal{R}/\delta)$$

$$\leq - \sum_{i=1}^n (P(x_i, a_i^1, a_i^2) - P^*(x_i, a_i^1, a_i^2))^2 + \log(N_\mathcal{R}/\delta),$$

where the second inequality uses Equation (13), and the last inequality uses Equation (14). By taking $P$ as $P_\theta$, since $r_\theta$ is the MLE which means $P_\theta$ also the MLE result, we get

$$\sum_{i=1}^{n}(P_\theta(x_i, a_i^1, a_i^2) - P^*(x_i, a_i^1, a_i^2))^2 \leq \sum_{i=1}^{n} \log \frac{P^*(y_i|x_i, a_i^1, a_i^2)}{P_\theta(y_i|x_i, a_i^1, a_i^2)} + \log(N_\mathcal{R}/\delta)$$
$$\leq \log(N_\mathcal{R}/\delta),$$

which concludes the first part, and we immediately get

$$\sum_{i=1}^{n} \left[\sigma\left(r_\theta(x_i, a_i^1) - r_\theta(x_i, a_i^2)\right) - \sigma\left(r^*(x_i, a_i^1) - r^*(x_i, a_i^2)\right)\right]^2 \leq \log \frac{N_\mathcal{R}}{\delta}.$$

With the fact that $\sigma'(r) = \sigma(r)(1 - \sigma(r)) \geq \frac{1}{2e}$ (as the reward is assumed to be bounded in $[0, 1]$), it can be established that

$$\sum_{i=1}^{n} \left[r_\theta(x_i, a_i^1) - r_\theta(x_i, a_i^2) - \left(r^*(x_i, a_i^1) - r^*(x_i, a_i^2)\right)\right]^2$$
$$\leq 2e \sum_{i=1}^{n} \left[\sigma(r_\theta(x_i, a_i^1) - r_\theta(x_i, a_i^2)) - \sigma\left(r^*(x_i, a_i^1) - r^*(x_i, a_i^2)\right)\right]^2$$
$$\leq 2e \cdot \log \frac{N_\mathcal{R}}{\delta},$$

which concludes the proof. □

**Lemma D.2.** *Consider arbitrary policies $\pi^1, \pi^2$, and a set of context-action pairs $\{(x_i, a_i^1, a_i^2, y_i)\}_{i=1}^{n}$ generated i.i.d. from the BT model where $a_i^1 \sim \pi^1, a_i^2 \sim \pi^2$. Suppose that $r_\theta$ is the MLE estimator. With probability at least $1 - \delta$, we have*

$$\mathbb{E}_{x \sim \rho, a^1 \sim \pi^1, a^2 \sim \pi^2}\left[\left(r_\theta(x, a^1) - r_\theta(x, a^2) - (r^*(x, a^1) - r^*(x, a^2))\right)^2\right] \leq \frac{6e}{n} \log\left(\frac{N_\mathcal{R}}{\delta}\right).$$

*Proof.* By the multiplicative Chernoff bounds (refer to Lemma H.4 and Remark H.5), with probability at least $1 - \delta$, for any $r \in \mathcal{R}_\Theta$, we have

$$\frac{n}{2}\mathbb{E}_{x \sim \rho, a^1 \sim \pi^1, a^2 \sim \pi^2}\left[\left(r(x, a^1) - r(x, a^2) - (r^*(x, a^1) - r^*(x, a^2))\right)^2\right]$$
$$\leq \sum_{i=1}^{n}\left(r(x_i, a_i^1) - r(x_i, a_i^2) - (r^*(x_i, a_i^1) - r^*(x_i, a_i^2))\right)^2 + \log\left(\frac{N_\mathcal{R}}{\delta}\right).$$

By taking $r = r_\theta$ and using Lemma D.1, we can get

$$\frac{n}{2}\mathbb{E}_{x \sim \rho, a^1 \sim \pi^1, a^2 \sim \pi^2}\left[\left(r_\theta(x, a^1) - r_\theta(x, a^2) - (r^*(x, a^1) - r^*(x, a^2))\right)^2\right]$$
$$\leq \sum_{i=1}^{n}\left(r_\theta(x_i, a_i^1) - r_\theta(x_i, a_i^2) - (r^*(x_i, a_i^1) - r^*(x_i, a_i^2))\right)^2 + \log\left(\frac{N_\mathcal{R}}{\delta}\right)$$
$$\leq (2e + 1)\log \frac{N_\mathcal{R}}{\delta} \leq 3e \log \frac{N_\mathcal{R}}{\delta},$$

which proves the lemma. □

For the bonus $b_t$, we choose $b_t(x, a^1, a^2) = \min\{1, \beta_T \cdot U(\xi, x, a^1, a^2; \mathcal{R}_t, \mathcal{D}_t)\}$ and

$$\mathcal{R}_t = \{r_\theta \in \mathcal{R}_\Theta : \sum_{i=1}^{t}(r_{\theta_t}(x_i, a_i^1) - r_{\theta_t}(x_i, a_i^2) - (r_\theta(x_i, a_i^1) - r_\theta(x_i, a_i^2)))^2 + \xi \leq \beta_T^2\},$$

where $\beta_T^2 = 4e \log(N_\mathcal{R}T/\delta)$ and $\xi \leq \beta_T^2/2$. The following lemma shows that $b_t$ is a valid bonus.

**Lemma D.3.** *Under Algorithm 1, we have with probability at least $1 - \delta$ for all $t \in [T]$, the uniform optimism event that*

$$\mathcal{E}_t = \{r_{\theta_t}(x, a^1) - r_{\theta_t}(x, a^2) + b_t(x, a^1, a^2) - (r^*(x, a^1) - r^*(x, a^2)) > 0, \quad \forall (x, a^1, a^2) \in \mathcal{X} \times \mathcal{A} \times \mathcal{A}\}$$

*holds true.*

*Proof.* By Lemma D.1, for all $t \in [T]$, with probability at least $1 - \delta$,

$$\sum_{i=1}^{t} \left[ r_{\theta_t}(x_i, a_i^1) - r_{\theta_t}(x_i, a_i^2) - \left( r^*(x_i, a_i^1) - r^*(x_i, a_i^2) \right) \right]^2 \leq 2e \log \frac{N_{\mathcal{R}} T}{\delta} = \frac{1}{2} \beta_T^2.$$

Hence, we deduce that for any $(x, a^1, a^2) \in \mathcal{X} \times \mathcal{A} \times \mathcal{A}$,

$$|r_{\theta_t}(x, a^1) - r_{\theta_t}(x, a^2) - (r^*(x, a^1) - r^*(x, a^2))|$$

$$\leq \sup_{R_1, R_2 \in \mathcal{R}_\Theta} \frac{|R_1(x, a^1) - R_1(x, a^2) - R_2(x, a^1) + R_2(x, a^2)|}{\sqrt{\xi + \sum_{i=1}^{t} (R_1(x_i, a_i^1) - R_1(x_i, a_i^2) - R_2(x_i, a_i^1) + R_2(x_i, a_i^2))^2}}$$

$$\cdot \sqrt{\xi + \sum_{i=1}^{t} (r_{\theta_t}(x_i, a_i^1) - r_{\theta_t}(x_i, a_i^2) - (r^*(x_i, a_i^1) - r^*(x_i, a_i^2)))^2}$$

$$\leq U(\xi, x, a^1, a^2; \mathcal{R}_t, \mathcal{D}_t) \sqrt{\xi + \frac{1}{2} \beta_T^2}$$

$$\leq U(\xi, x, a^1, a^2; \mathcal{R}_t, \mathcal{D}_t) \beta_T,$$

which concludes the proof. $\qquad \square$

Based on previous decomposition lemma (Lemma C.6), we can get the following bound for $J_f(\pi_f^*) - J_f(\pi_t)$ in Algorithm 1.

**Lemma D.4** (Objective Decomposition). *For $t \in [T]$, conditioning on the uniform optimism event that $\mathcal{E}_t$ holds, we have*

$$J_f(\pi_f^*) - J_f(\pi_t) \leq 4\eta \mathcal{C}(f, \mathcal{R}_\Theta, \eta) \mathbb{E}_{a^1 \sim \pi_t, a^2 \sim \pi_{t-1}}[(b_{t-1}(x, a^1, a^2))^2].$$

*Proof.* Denote

$$\tilde{r}_t(x, a) = \hat{r}_t(x, a) - \mathbb{E}_{a' \sim \pi_t(\cdot|x)}[r(x, a') - r^*(x, a')]$$

$$= r_{\theta_t}(x, a) + \mathbb{E}_{a' \sim \pi_t(\cdot|x)}[b_t(x, a, a')] - \mathbb{E}_{a' \sim \pi_t(\cdot|x)}[r(x, a') - r^*(x, a')].$$

By Lemma C.4, $J_f(\pi_{\hat{r}_t}) = J_f(\pi_{\tilde{r}_t})$. Note that

$$r_{\theta_t}(x, a) + \mathbb{E}_{a' \sim \pi_t(\cdot|x)}[b_t(x, a, a')] - [r(x, a') - r^*(x, a')]] - r^*(x, a) \geq 2\mathbb{E}_{a' \sim \pi_t}[b_t(x, a, a')] \geq 0.$$

Using the above result in Lemma C.6, we have

$$J_f(\pi^*) - J_f(\pi_t) = J_f(\pi^*) - J_f(\pi_{\tilde{r}_t})$$

$$\leq 4\eta \mathcal{C}(f, \mathcal{R}_\Theta, \eta) \mathbb{E}_{a^1 \sim \pi_t, a^2 \sim \pi_{t-1}}[(b_{t-1}(x, a^1, a^2))^2],$$

which concludes the proof. $\qquad \square$

With all the above results, we can prove Theorem 3.3.

*Proof of Theorem 3.3.* By Lemma D.4, we can bound the regret as:

$$\text{Regret}_f(T) = \sum_{t=1}^{T} J_f(\pi_f^*) - J_f(\pi_t)$$

$$\leq 4\eta \mathcal{C}(f, \mathcal{R}_\Theta, \eta) \sum_{t=1}^{T} \mathbb{E}_{a^1 \sim \pi_t, a^2 \sim \pi_{t-1}}[(b_{t-1}(x, a^1, a^2))^2]$$

$$= 4\eta \mathcal{C}(f, \mathcal{R}_\Theta, \eta)\beta_T^2 \sum_{t=1}^{T} \mathbb{E}_{x_t \sim \rho, a_t^1 \sim \pi_t, a_t^2 \sim \pi_{t-1}} \min\left\{1, \left(U(\xi, x_t, a_t^1, a_t^2; \mathcal{R}_{t-1}, \mathcal{D}_{t-1})\right)^2\right\}$$

$$= \mathcal{O}\left(\eta \mathcal{C}(f, \mathcal{R}_\Theta, \eta) \log(N_{\mathcal{R}}T/\delta)d(\mathcal{R}_\Theta, \xi, T)\right),$$

which concludes the proof. □

## E. Proof of Theorem 4.3

In this section, we present the proof of Theorem 4.3. Before that, we establish some propositions about our algorithm design. Let $\mu(a^1, a^2|x)$ represent the probability of choosing $a^1$ as the first action and $a^2$ as the second action, and we have

$$\mu(a^1, a^2|x) = (1 - p(x))\pi_\theta'(a^1|x)\pi_\theta'(a^2|x) + p(x)\pi_\theta^+(a^1|x)\pi_\theta^-(a^2|x)$$

$$= \frac{\pi_\theta'(a^1|x)\pi_\theta'(a^2|x)}{1 + Z_\theta^+(x)Z_\theta^-(x)} \left\{1 + \exp\{r_\theta(x, a^1) - r_\theta(x, a^2)\}\right\}.$$

Let $\overline{\mu}(a^1, a^2|x)$ be the probability that the sampled two actions are $a^1, a^2$, and we can get

$$\overline{\mu}(a^1, a^2|x) = \frac{1}{2}\left(\mu(a^1, a^2|x) + \mu(a^2, a^1|x)\right).$$

The ratio between $\overline{\mu}$ and $\pi_\theta'$ will be

$$\frac{\overline{\mu}(a^1, a^2|x)}{\pi_\theta'(a^1|x)\pi_\theta'(a^2|x)} = \frac{\pi_\theta'(a^1|x)\pi_\theta'(a^2|x)}{2 + 2Z_\theta^+(x)Z_\theta^-(x)}\left\{2 + \exp\{r_\theta(x, a^1) - r_\theta(x, a^2)\} + \exp\{r_\theta(x, a^2) - r_\theta(x, a^1)\}\right\}$$

$$= \frac{\pi_\theta'(a^1|x)\pi_\theta'(a^2|x)}{2 + 2Z_\theta^+(x)Z_\theta^-(x)} \frac{1}{\sigma'(r_\theta(x, a^1) - r_\theta(x, a^2))}$$

$$= \frac{\pi_\theta'(a^1|x)\pi_\theta'(a^2|x)}{2 + 2Z_\theta^+(x)Z_\theta^-(x)} \frac{1}{\text{Var}_{r_\theta}(\mathbf{1}\{a^1 = a^\omega\}|x, a^1, a^2)}, \tag{15}$$

where the variance term is

$$\text{Var}_{r_\theta}(\mathbf{1}\{a^1 = a^\omega\}|x, a^1, a^2) = \sigma(r_\theta(x, a^2) - r_\theta(x, a^1))\sigma(r_\theta(x, a^1) - r_\theta(x, a^2)).$$

To establish the result, we impose the following regularity condition. There exists a constant $C \geq 1$ satisfying

$$\text{Var}_{r_\theta}(\mathbf{1}\{a^1 = a^\omega\}|x, a^1, a^2) \leq C \cdot \text{Var}_{r^*}(\mathbf{1}\{a^1 = a^\omega\}|x, a^1, a^2) \tag{16}$$

for any context $x \in \mathcal{X}$ and action $a^1, a^2$, Here $\text{Var}_{r_\theta}(\mathbf{1}\{a^1 = a^\omega\}|x, a^1, a^2)$ denotes the conditional variance under the BT model, when the implicit reward function $r^*$ is replaced by $r_\theta$. The term $\text{Var}_{r^*}(\mathbf{1}\{a^1 = a^\omega\}|x, a^1, a^2) = \text{Var}(\mathbf{1}\{a^1 = a^\omega\}|x, a^1, a^2)$ represents the conditional variance under the ground-truth reward.

Recall that the modified loss function is

$$\mathcal{L}(\theta) := -\frac{1}{n}\sum_{i=1}^{n} \omega(x_i) \log \sigma(r_\theta(x_i, a_i^\omega) - r_\theta(x_i, a_i^l)),$$

and we denote $\boldsymbol{\omega} = \{\omega(x_i)\}_{i=1}^{n}$. With the sampling strategy designed in Algorithm 2, $\hat{\theta}$ has the following approximation result.

**Theorem E.1.** *Assume the reward model* $r^* = r_{\theta^*}$, *for some* $\theta^*$, *under mild regularity condition, the estimated* $\hat{\theta}$ *asymptotically follows a Gaussian distribution:*

$$\sqrt{n}(\hat{\theta} - \theta^*) \xrightarrow{d} \mathcal{N}(0, \boldsymbol{\Omega}) \qquad n \to \infty.$$

The covariance $\boldsymbol{\Omega}$ can be bounded by $\boldsymbol{\Omega} \preceq ||\boldsymbol{\omega}||_\infty \cdot \boldsymbol{\Sigma}_*^{-1}$. When using the above Algorithm 2, we further have

$$\boldsymbol{\Sigma}_* \succeq \frac{1}{C\overline{Z}_\phi} \mathbb{E}_{x\sim\rho}\left[\overline{T}_{\theta^*}(x)\mathrm{Cov}_{a\sim\pi_*'(\cdot|x)}\left[\nabla_\theta r^*(x,a)|x\right]\right].$$

*Proof of Theorem E.1.* By Theorem 4.4 in Feng et al. (2025), we have

$$\sqrt{n}(\hat{\theta} - \theta^*) \xrightarrow{d} \mathcal{N}(0, \boldsymbol{\Omega}), \quad \boldsymbol{\Omega} \preceq ||\boldsymbol{\omega}||_\infty \cdot \boldsymbol{\Sigma}_*^{-1}$$

where

$$\boldsymbol{\Sigma}_* = \mathbb{E}_{x\sim\rho,(a^1,a^2)\sim\overline{\mu}(\cdot|x)}[\omega(x)\mathrm{Var}(\mathbf{1}\{a^1 = a^\omega\}|x,a^1,a^2)\cdot\boldsymbol{g}\boldsymbol{g}^\top],$$

and

$$\mathrm{Var}(\mathbf{1}\{a^1 = a^\omega\}|x,a^1,a^2) = \sigma(r^*(x,a^1) - r^*(x,a^2))\sigma(r^*(x,a^2) - r^*(x,a^1)),$$
$$\boldsymbol{g} = \nabla_\theta r^*(x,a^1) - \nabla_\theta r^*(x,a^2).$$

Recall that

$$\omega(x) := \{\overline{T}_\theta(x) + Z_\theta^+(x)Z_\theta^-(x)\overline{T}_\theta(x)\}/\overline{Z}_\theta.$$

Substitute $\overline{\mu}$ in Equation (15) in to $\boldsymbol{\Sigma}_*$, we have

$$\boldsymbol{\Sigma}_* = \mathbb{E}_{x\sim\rho,(a^1,a^2)\sim\overline{\mu}(\cdot|x)}[\omega(x)\mathrm{Var}(\mathbf{1}\{a^1 = a^\omega\}|x,a^1,a^2)\cdot\boldsymbol{g}\boldsymbol{g}^\top]$$
$$= \mathbb{E}_{x\sim\rho,(a^1,a^2)\sim\pi_\theta'(\cdot|x)}\left[\frac{\overline{\mu}(a^1,a^2|x)}{\pi_\theta'(a^1|x)\pi_\theta'(a^2|x)}\omega(x)\mathrm{Var}(\mathbf{1}\{a^1 = a^\omega\}|x,a^1,a^2)\cdot\boldsymbol{g}\boldsymbol{g}^\top\right]$$
$$\succeq \frac{1}{2C\overline{Z}_\theta}\mathbb{E}_{x\sim\rho}\overline{T}_\theta(x)\mathbb{E}_{(a^1,a^2)\sim\pi_\theta'(\cdot|x)}[\boldsymbol{g}\boldsymbol{g}^\top]$$
$$= \frac{1}{C\overline{Z}_\theta}\mathbb{E}_{x\sim\rho}\left[\overline{T}_\theta(x)\mathrm{Cov}_{a\sim\pi_\theta'(\cdot|x)}\left[\nabla_\theta r^*(x,a)|x\right]\right],$$

where the inequality is from Equation (16) and the last equality is from

$$\mathbb{E}_{(a^1,a^2)\sim\pi_\theta'(\cdot|x)}[\boldsymbol{g}\boldsymbol{g}^\top|x]$$
$$= \mathbb{E}_{(a^1,a^2)\sim\pi_\theta'(\cdot|x)}[(\nabla_\theta r^*(x,a^1) - \nabla_\theta r^*(x,a^2))(\nabla_\theta r^*(x,a^1) - \nabla_\theta r^*(x,a^2))^\top|x]$$
$$= 2\cdot\mathbb{E}_{a\sim\pi_\theta'(\cdot|x)}[\nabla_\theta r^*(x,a)\nabla_\theta r^*(x,a)^\top|x]$$
$$\qquad - 2\cdot\mathbb{E}_{a\sim\pi_\theta'(\cdot|x)}[\nabla_\theta r^*(x,a)|x]\cdot\mathbb{E}_{a\sim\pi_\theta'(\cdot|x)}[\nabla_\theta r^*(x,a)|x]^\top$$
$$= \mathrm{Cov}_{a\sim\pi_\theta'(\cdot|x)}\left[\nabla_\theta r^*(x,a)|x\right].$$

Thus, when $\theta = \theta^*$, we have

$$\boldsymbol{\Sigma}_* \succeq \frac{1}{C\overline{Z}_{\theta^*}}\mathbb{E}_{x\sim\rho}\left[\overline{T}_\theta(x)\mathrm{Cov}_{a\sim\pi_*'(\cdot|x)}\left[\nabla_\theta r^*(x,a)|x\right]\right].$$

$\square$

Before digging deeper into the gradient of the value function $J_f(\pi_\theta)$, we denote $h = (f')^{-1}$, and we have

$$\pi_\theta(a|x) = \pi_0(a|x)h(\eta(r_\theta(x,a) - \lambda_\theta(x))).$$

Noticing that

$$\sum_a \pi_0(a|x)h(\eta(r_\theta(x,a) - \lambda_\theta(x))) = 1,$$

we have

$$\sum_a \pi_0(a|x)h'(\eta(r_\theta(x,a) - \lambda_\theta(x)))(\nabla_\theta r_\theta(x,a) - \nabla_\theta\lambda_\theta(x))\eta = 0,$$

which means

$$\nabla_\theta\lambda_\theta(x) = \frac{\sum_a T_\theta(x,a) \cdot \nabla_\theta r_\theta(x,a)}{\sum_a T_\theta(x,a)},$$

where

$$T_\theta(x,a) = \pi_0(x,a)h'(\eta(r_\theta(x,a) - \lambda_\theta(x))), \quad \overline{T}_\theta(x) = \sum_a T_\theta(x,a).$$

Now, we begin to prove Theorem 4.3 by showing the Hessian matrix of the $J_f(\pi_\theta)$ at $\theta = \theta^*$ is $-\eta\Sigma_*^1$, i.e., $\nabla_\theta^2 J_f(\pi_\theta)|_{\theta=\theta^*} = -\eta\Sigma_*^1$, where

$$\Sigma_*^1 = \mathbb{E}_{x\sim\rho}\left[\overline{T}_\theta(x)\text{Cov}_{a\sim\pi'_\theta(\cdot|x)}[\nabla_\theta r_\theta(x,a)]|_{\theta=\theta^*}\right].$$

*Proof for Theorem 4.3.* For $\nabla_\theta(\pi_\theta(a|x))$, we have

$$\nabla_\theta(\pi_\theta(a|x)) = \pi_0(a|x)h'(\eta(r_\theta(x,a) - \lambda_\theta(x))) \cdot (\eta\nabla_\theta r_\theta(x,a) - \eta\nabla_\theta\lambda_\theta(x))$$
$$= \eta\pi_0(a|x)h'(\eta(r_\theta(x,a) - \lambda_\theta(x))) \cdot (\nabla_\theta r_\theta(x,a) - \mathbb{E}_{a\sim\pi'_\theta(\cdot|x)}[\nabla_\theta r_\theta(x,a)]).$$

Also

$$\sum_a \lambda_\theta(x)\nabla_\theta(\pi_\theta(a|x)) = \lambda_\theta(x)\sum_a \nabla_\theta(\pi_\theta(a|x))$$
$$= \sum_a T_\theta(x,a)\nabla_\theta r_\theta(x,a) - \sum_a T_\theta(x,a) \cdot \mathbb{E}_{a\sim\pi'_\theta(\cdot|x)}[\nabla_\theta r_\theta(x,a)])$$
$$= 0,$$

where $\pi'_\theta(a|x) = T_\theta(x,a)/\sum_a T_\theta(x,a)$.

Now, we compute $\nabla_\theta(D_f(\pi_\theta, \pi_0|x))$ and $\nabla_\theta\mathbb{E}_{a\sim\pi_\theta(\cdot|x)}[r^*(x,a)]$.

$$\nabla_\theta(D_f(\pi_\theta, \pi_0|x)) = \mathbb{E}_{a\sim\pi_0(\cdot|x)}[f'(\frac{\pi_\theta(a|x)}{\pi_0(a|x)})\nabla_\theta(\frac{\pi_\theta(a|x)}{\pi_0(a|x)})]$$
$$= \mathbb{E}_{a\sim\pi_0(\cdot|x)}[\eta(r_\theta(x,a) - \lambda_\theta(x))\nabla_\theta(\frac{\pi_\theta(a|x)}{\pi_0(a|x)})]$$
$$= \mathbb{E}_{a\sim\pi_0(\cdot|x)}[\eta(r_\theta(x,a) - \lambda_\theta(x))\frac{\nabla_\theta\pi_\theta(a|x)}{\pi_0(a|x)}]$$
$$= \mathbb{E}_{a\sim\pi_0(\cdot|x)}[\eta(r_\theta(x,a))\frac{\nabla_\theta\pi_\theta(a|x)}{\pi_0(a|x)}]$$
$$= \eta^2\overline{T}_\theta(x)\mathbb{E}_{a\sim\pi'_\theta(\cdot|x)}[(r_\theta(x,a))(\nabla_\theta r_\theta(x,a) - \mathbb{E}_{a\sim\pi'_\theta(\cdot|x)}[\nabla_\theta r_\theta(x,a)])],$$

where

$$\overline{T}_\theta(x) = \sum_a T_\theta(x,a) = \sum_a \pi_0(a|x)h'(\eta(r_\theta(a|x) - \lambda_\theta(x)))$$

and

$$\nabla_\theta \mathbb{E}_{a \sim \pi_\theta(\cdot|x)}[r^*(x,a)] = \nabla_\theta \sum_a \pi_\theta(a|x)[r^*(x,a)]$$

$$= \sum_a (\nabla_\theta \pi_\theta(a|x))[r^*(x,a)]$$

$$= \eta \overline{T}_\theta(x) \mathbb{E}_{a \sim \pi'_\theta(\cdot|x)}[r^*(x,a)(\nabla_\theta r_\theta(x,a) - \mathbb{E}_{a \sim \pi'_\theta(\cdot|x)}[\nabla_\theta r_\theta(x,a)])].$$

Combining above formulas, we get

$$\nabla_\theta J_f(\pi_\theta)$$

$$= \nabla_\theta \mathbb{E}_{x \sim \rho}[\mathbb{E}_{a \sim \pi_\theta(\cdot|x)}[r^*(x,a)] - \eta^{-1} \nabla_\theta(D_f(\pi_\theta, \pi_0|x))]$$

$$= \eta \mathbb{E}_{x \sim \rho}\left[\overline{T}_\theta(x) \mathbb{E}_{a \sim \pi'_\theta(\cdot|x)}[(r^*(x,a) - r_\theta(x,a))(\nabla_\theta r_\theta(x,a) - \mathbb{E}_{a \sim \pi'_\theta(\cdot|x)}[\nabla_\theta r_\theta(x,a)])]\right]$$

$$= \eta \mathbb{E}_{x \sim \rho}\left[\sum_a T_\theta(x,a)[(r^*(a) - r_\theta(a))(\nabla_\theta r_\theta(a) - \mathbb{E}_{a \sim \pi'_\theta(\cdot|x)}[\nabla_\theta r_\theta(a)])]\right].$$

For $\nabla^2_\theta J_f(\pi_\theta)$, we can use $\nabla_\theta$ on $(\nabla_\theta J_f(\pi_\theta))$ and we will get three parts $\nabla^2_\theta J_f(\pi_\theta) = \Gamma_1 + \Gamma_2 + \Gamma_3$ specifying as the following:

$$\Gamma_1 = \eta \mathbb{E}_{x \sim \rho}\left[\sum_a (r^*(x,a) - r_\theta(x,a))(\nabla_\theta T_\theta(x,a))(\nabla_\theta r_\theta(a) - \mathbb{E}_{a \sim \pi'_\theta}[\nabla_\theta r_\theta(x,a)])\right],$$

$$\Gamma_2 = -\eta \mathbb{E}_{x \sim \rho}\left[\overline{T}_\theta(x) \mathbb{E}_{a \sim \pi'_\theta}[(\nabla_\theta r_\theta(x,a) - \mathbb{E}_{a \sim \pi'_\theta}[\nabla_\theta r_\theta(x,a)])(\nabla_\theta r_\theta(x,a))^\top]\right],$$

$$\Gamma_3 = \mathbb{E}_{x \sim \rho}\left[\sum_a (r^*(x,a) - r_\theta(x,a))T_\theta(x,a)\nabla_\theta(\nabla_\theta r_\theta(x,a) - \mathbb{E}_{a \sim \pi'_\theta}[\nabla_\theta r_\theta(x,a)])\right].$$

where $\Gamma_1$ is the result using $\nabla_\theta$ on $T_\theta$ term, $\Gamma_2$ is the result using $\nabla_\theta$ on $r^* - r_\theta$ term, and $\Gamma_3$ is the result using $\nabla_\theta$ on $(\nabla_\theta r_\theta(x,a) - \mathbb{E}_{a \sim \pi'_\theta}[\nabla_\theta r_\theta(x,a)])$ term. Among those three terms, when $\theta = \theta^*$, we have $\Gamma_1 = \Gamma_3 = 0$ and

$$\Gamma_2 = -\eta \mathbb{E}_{x \sim \rho}\left[\overline{T}_\theta(x)\left[\mathbb{E}_{a \sim \pi'_\theta}[\nabla_\theta r_\theta(x,a) \cdot \nabla_\theta r_\theta(x,a)^\top] + \mathbb{E}_{a \sim \pi'_\theta}[\nabla_\theta r_\theta(x,a)]\mathbb{E}_{a \sim \pi'_\theta}[\nabla_\theta r_\theta(x,a)]^\top\right]\right]$$

$$= -\eta \mathbb{E}_{x \sim \rho}\left[\overline{T}_\theta(x)\text{Cov}_{a \sim \pi'_\theta(\cdot|x)}[\nabla_\theta r_\theta(x,a)]\right].$$

Thus, we conclude our proof. $\qquad \square$

Proposition 4.4 can be obtained by the above Hessian result. Before giving the proof to Proposition 2.3, we introduce a definition to measure the difference between $(f')^{-1}$ and its derivative under all possible reward functions $r_\theta$.

**Definition E.2.** For function $f$, let $h = (f')^{-1}$ we define

$$\mathcal{M}(f, \mathcal{R}_\Theta, \eta) := \max_{r \in \overline{\mathcal{R}_\Theta}} \max_x \sum_{a \in \mathcal{A}} \pi_0(a|x)h'(\eta(r(x,a) - \lambda_r(x))),$$

where $\lambda_\theta$ satisfies

$$\sum_a \pi_0(a|x)h(\eta(r_\theta(x,a) - \lambda_\theta(x))) = 1.$$

**Remark**: In the KL divergence case, $\mathcal{M}(f_{kl}, \mathcal{R}_\Theta, \pi_0) = 1, \forall \Theta, \pi_0$. $\mathcal{M}(f, \mathcal{R}_\Theta, \eta)$ acts similarly to $\mathcal{C}(f, \mathcal{R}_\Theta, \eta)$ (in Definition C.5). In reality, we have $\mathcal{C}(f, \mathcal{R}_\Theta, \eta) \geq \mathcal{M}(f, \mathcal{R}_\Theta, \eta)$ (Appendix F). Also this will be the upper bound for $\overline{T}_\theta$, i.e., $\overline{T}_\theta \leq \mathcal{M}(f, \mathcal{R}_\Theta, \eta)$.

*Proof of Proposition 4.4.* Since $\pi^*$ is the optimal policy, we have $\nabla_\theta J_f(\pi^*) = 0$. By Theorem 4.3, we have

$$
\begin{aligned}
J_f(\pi^*) - J_f(\pi_\theta) &= \frac{1}{2}(\hat{\theta} - \theta^*)^\top (-\nabla_\theta^2 J_f(\pi_\theta)|_{\theta=\theta^*})(\hat{\theta} - \theta^*) + o(||(\hat{\theta} - \theta^*)||_2^2) \\
&= \frac{1}{2}(\hat{\theta} - \theta^*)^\top (\eta \boldsymbol{\Sigma}_*^1)(\hat{\theta} - \theta^*) + o(||(\hat{\theta} - \theta^*)||_2^2).
\end{aligned}
\tag{17}
$$

By Theorem E.1, we have

$$
T \cdot \{J_f(\pi^*) - J_f(\hat{\pi})\} \xrightarrow{d} \frac{1}{2} z^\top \boldsymbol{\Omega}^{\frac{1}{2}} H \boldsymbol{\Omega}^{\frac{1}{2}} z := X,
$$

where $z \sim \mathcal{N}(0, I)$, $H = \eta \boldsymbol{\Sigma}_*^1$ and this asymptotic result will be rigorously proved in Lemma E.3. $\boldsymbol{\Omega}$ satisfies

$$
\boldsymbol{\Omega} \preceq C \overline{Z}_\theta ||\boldsymbol{\omega}||_\infty \cdot (\boldsymbol{\Sigma}_*^1)^{-1},
$$

and the matrix $\boldsymbol{\Omega}^{\frac{1}{2}} H \boldsymbol{\Omega}^{\frac{1}{2}}$ can be bounded by

$$
\boldsymbol{\Omega}^{\frac{1}{2}} H \boldsymbol{\Omega}^{\frac{1}{2}} \preceq C \overline{Z}_\theta ||\boldsymbol{\omega}||_\infty \cdot (\boldsymbol{\Sigma}_*^1)^{-\frac{1}{2}} H (\boldsymbol{\Sigma}_*^1)^{-\frac{1}{2}} = C \overline{Z}_\theta ||\boldsymbol{\omega}||_\infty \eta \cdot I.
$$

Thus

$$
X \leq 2\eta C \mathcal{M}(f, \mathcal{R}_\Theta, \eta)(||\overline{T}_\theta + \overline{T}_\theta Z_\theta^+ Z_\theta^-||_\infty) \cdot z^\top z,
$$

where $z^\top z$ follows a chi-squared distribution with $d$ degrees of freedom, and $X$ is stochastically dominated by a rescaled chi-squared random variable

$$
2\eta C \mathcal{M}(f, \mathcal{R}_\Theta, \eta)(||\overline{T}_\theta + \overline{T}_\theta Z_\theta^+ Z_\theta^-||_\infty) \cdot \chi_d^2.
$$

Equivalently, we can express this dominance as

$$
\limsup_{T \to \infty} \mathbb{P} \left\{ T(J_f(\pi^*) - J_f(\hat{\pi})) \leq 2\eta C \mathcal{M}(f, \mathcal{R}_\Theta, \eta)(||\overline{T}_\theta + \overline{T}_\theta Z_\theta^+ Z_\theta^-||_\infty) \cdot t \right\} \leq \mathbb{P}\{\chi_d \geq t\}.
$$

Applying the tail bound in Lemma H.8, we can get

$$
\begin{aligned}
\limsup_{T \to \infty} \mathbb{P} \left\{ J_f(\pi^*) \leq J_f(\hat{\pi}) - C_0 \cdot \frac{d(1 + \varepsilon)}{T} \right\} &\leq \mathbb{P} \left\{ \chi_d^2 \geq d(1 + \varepsilon) \right\} \\
&\leq \exp\left( -\frac{d}{2}(\varepsilon - \log(1 + \varepsilon)) \right),
\end{aligned}
$$

where $C_0 = 2\eta C \mathcal{M}(f, \mathcal{R}_\Theta, \eta)(||\overline{T}_{\theta^*} + \overline{T}_{\theta^*} Z_{\theta^*}^+ Z_{\theta^*}^-||_\infty)$. $\qquad\square$

**Lemma E.3.** *The value function satisfies*

$$
n \cdot \{J_f(\pi^*) - J_f(\hat{\pi})\} \xrightarrow{d} \frac{1}{2} z^\top \boldsymbol{\Omega}^{\frac{1}{2}} H \boldsymbol{\Omega}^{\frac{1}{2}} z.
$$

*Proof.* The proof for this lemma is given in Feng et al. (2025), and we include the proof here for completeness. To prove this lemma, we first recast the value gap into the product of two terms and then invoke Slutsky's theorem.

$$
n \cdot (J_f(\pi^*) - J_f(\pi_\theta)) = \underbrace{n \cdot (\theta - \theta^*)^\top H (\theta - \theta^*)}_{U_n} \cdot \underbrace{\frac{J_f(\pi^*) - J_f(\pi_\theta)}{(\theta - \theta^*)^\top H (\theta - \theta^*)}}_{V_n}.
\tag{18}
$$

By isolating $U_n$ and $V_n$ in this way, we can handle their limiting behaviors separately. We will prove that

$$
U_n \xrightarrow{d} z^\top \boldsymbol{\Omega}^{\frac{1}{2}} H \boldsymbol{\Omega}^{\frac{1}{2}} z \qquad \text{with } z \sim \mathcal{N}(0, I),
\tag{19a}
$$

$$
V_n \xrightarrow{p} \frac{1}{2}.
\tag{19b}
$$

If these two results are established, the desired asymptotic distribution of the value gap follows directly from Slutsky's theorem.

To complete the proof, we verify Equations (19a) and (19b). Equation (19a) is a straightforward corollary of Theorem E.1, and we proof Equation (19b) below.

Since $\boldsymbol{\Sigma}_*^1$ is non-singular, the matrix $H = (\eta \overline{T}_\theta) \cdot \boldsymbol{\Sigma}_*^1$ is also non-singular. From Equation (17), we know that for any $\varepsilon \in (0,1)$, there exists a threshold $\epsilon(\varepsilon) > 0$ such that whenever $||\theta - \theta^*||_2 \leq \eta(\varepsilon)$, the following inequality holds:

$$\left(\frac{1}{2} - \varepsilon\right)(\theta - \theta^*)^\top H (\theta - \theta^*) \ \leq \ J_f(\pi^*) - J_f(\pi_\theta) \ \leq \ \left(\frac{1}{2} + \varepsilon\right)(\theta - \theta^*)^\top H (\theta - \theta^*).$$

This can be reformulated as

$$\left| V_n - \frac{1}{2} \right| \ \leq \ \varepsilon.$$

Next, under the condition that $\hat{\theta} \xrightarrow{p} \theta^*$, for any $\delta > 0$, there exists an integer $N(\varepsilon, \delta) \in \mathbb{Z}_+$ such that for any $n \geq N(\varepsilon, \delta)$,

$$\mathbb{P}\big(||\hat{\theta} - \theta^*||_2 \geq \epsilon(\varepsilon)\big) \leq \delta.$$

Therefore, for any $n \geq N(\varepsilon, \delta)$, we can conclude

$$\mathbb{P}\left(\left| V_n - \frac{1}{2} \right| \geq \varepsilon\right) \ \leq \ \delta.$$

In simpler terms, $V_n \xrightarrow{p} \frac{1}{2}$, which establishes Equation (19b). $\qquad\square$

# F. Factors $\mathcal{C}$ and $\mathcal{M}$ under Different Divergences

In this section, we prove the following results mentioned in Sec. 5:

- With $f = x \log x$ (i.e., KL divergence), it holds that $\mathcal{C}(f, \mathcal{R}_\Theta, \eta) = \mathcal{M}(f, \mathcal{R}_\Theta, \eta) = 1$;

- With $f = x \log x + (x-1)^2$ (i.e., chi-squared mixed KL) or $f = x \log x - \log x$, it holds that $\mathcal{M}(f, \mathcal{R}_\Theta, \eta) \leq \mathcal{C}(f, \mathcal{R}_\Theta, \eta) < 1$;

- With $f = -\log x$ (i.e., forward-KL), it holds that $\mathcal{C}(f, \mathcal{R}_\Theta, \eta) \geq \mathcal{M}(f, \mathcal{R}_\Theta, \eta) > 1$;

*Proof.* First, we prove that $\mathcal{C}(f, \mathcal{R}_\Theta, \eta) \geq \mathcal{M}(f, \mathcal{R}_\Theta, \eta)$ for any $f, \mathcal{R}_\Theta, \eta$. Recall that

$$\mathcal{C}(f, \mathcal{R}_\Theta, \eta) := \max_{r \in \overline{\mathcal{R}_\Theta}} \max_{x,a} \frac{h'(\eta(r(x,a)) - \lambda_r(x))}{h(\eta(r(x,a)) - \lambda_r(x))}$$

$$\mathcal{M}(f, \mathcal{R}_\Theta, \eta) := \max_{r \in \overline{\mathcal{R}_\Theta}} \max_{x} \sum_{a \in \mathcal{A}} \pi_0(a|x) h'(\eta(r(x,a) - \lambda_r(x))).$$

Then, we have

$$\frac{\mathcal{M}(f, \mathcal{R}_\Theta, \eta)}{\mathcal{C}(f, \mathcal{R}_\Theta, \eta)} \leq \max_{r,x} \sum_{a \in \mathcal{A}} \pi_0(a|x) h'(\eta(r(x,a) - \lambda_r(x))) \cdot \frac{h(\eta(r(x,a)) - \lambda_r(x))}{h'(\eta(r(x,a)) - \lambda_r(x))}$$

$$\leq \max_{r,x} \sum_{a \in \mathcal{A}} \pi_0(a|x) h(\eta(r(x,a) - \lambda_r(x))) = 1,$$

which indicates $\mathcal{C}(f, \mathcal{R}_\Theta, \eta) \geq \mathcal{M}(f, \mathcal{R}_\Theta, \eta)$.

In the following, we provide further bounds for specific divergence choices:

- $f(x) = x \log x$ (KL divergence): It can be clearly observed that $h(y) = h'(y) = \exp(y - 1)$ correspondingly, which directly leads to $\mathcal{C}(f, \mathcal{R}_\Theta, \eta) = \mathcal{M}(f, \mathcal{R}_\Theta, \eta) = 1$.

- $f(x) = x \log x + (x-1)^2$ (Chi-squared mixed KL): We have $f'(x) = \log x + 2x - 1$, $h(y) = f'^{-1}(y)$ and

$$h'(y) = \frac{1}{f''(h(y))} = \frac{1}{\frac{1}{h(y)} + 2} = \frac{h(y)}{1 + 2h(y)}.$$

Thus, we have

$$\frac{h'(y)}{h(y)} = \frac{1}{1 + 2h(y)} < 1, \quad \forall y$$

since $h(y) > 0$. We have $\mathcal{C}(f, \mathcal{R}_\Theta, \eta) < 1$.

- $f(x) = x \log x - \log x$: In this case, we have $f'(x) = \log x + 1 - \frac{1}{x}$, $h(y) = f'^{-1}(y)$ and

$$h'(y) = \frac{1}{f''(h(y))} = \frac{h(y)}{1 + \frac{1}{h(y)}}.$$

Thus, we have

$$\frac{h'(y)}{h(y)} = \frac{1}{1 + \frac{1}{h(y)}} < 1, \quad \forall y$$

by the fact that $h(y) > 0$. Therefore we get $\mathcal{C}(f, \mathcal{R}_\Theta, \eta) < 1$.

- $f(x) = -\log x$ (forward-KL): We have $f'(x) = -\frac{1}{x}$ and $h(y) = f'^{-1}(y) = -\frac{1}{x}$.

Notice that $h'(y) = \frac{1}{y^2} = (h(y))^2$ and

$$
1 = \left( \sum_a \pi_0(a|x) h(\eta(r(x,a) - \lambda_r(x))) \right)^2 \le \left( \sum_a \pi_0(a|x) \right) \cdot \left( \sum_a \pi_0(a|x) h^2(\eta(r(x,a) - \lambda_r(x))) \right)
$$
$$
= \sum_a \pi_0(a|x) h'(\eta(r(x,a) - \lambda_r(x)))
$$
$$
\le \mathcal{M}(f, \mathcal{R}_\Theta, \eta),
$$

which concludes the proof. $\square$

## G. Experiment Details

### G.1. Implementation Considerations of Computing the Optimal Policy

Although the optimal policy $\pi_\theta$ is characterized in Proposition 2.3, to obtain the value of $\lambda_\theta$, it still requires us to solve the following equation for a given context $x$:

$$\sum_a \pi_0(a|x) h(\eta(r_\theta(x,a) - \lambda_\theta)) = 1.$$

For a general $f$, the above equation does not admit an explicit solution for $\lambda_\theta(x)$. We chose to solve this equation in Python using the "numpy.nsolve()" function to find a numerical solution for $\lambda_\theta(x)$, and use it to compute the numerical optimal policy $\pi_\theta(a|x) = \pi_0(a|x) h(\eta(r_\theta(x,a) - \hat{\lambda}_\theta))$.

### G.2. Experiment Setup

We limit the scope to the linear case and assume the action and context are vectors in $\mathbb{R}^k$. In the Bradley-Terry model, we parameterize the reward function $R^* \in \mathcal{R}$ by a matrix $W$ (with size $k \times k$). We give the exact form below.

**Bradley-Terry model.** We parameterize the reward function $R \in \mathcal{R}$ by a matrix $W$ (with size $k \times k$) as

$$R : \mathcal{X} \times \mathcal{A} \to [0, 1], \quad (x, a) \mapsto x^T W a.$$

For implementation, we choose $k = 5$ and first uniformly randomly sample from $[0, 1]$ to construct the ground-truth preference model parameters $M^*$ and $W^*$. We similarly sample 10 vectors from $[0, 1]^5$ as the action set $\mathcal{A}$. In each iteration, we randomly sample a vector from the uniform distribution in $[0, 1]^5$ as the context vector, and then we sample action pairs $(a^1, a^2)$ based on the respective sampling strategy. We run the trajectory for $T$ iterations and repeat the experiments 5 times, computing the averages and standard deviations.

## H. Auxiliary Lemmas

**Lemma H.1.** *For any random variable $X \geq 0$, we have*

$$\mathbb{E}[X^3] - \mathbb{E}[X^2]\mathbb{E}[X] \geq 0.$$

*Proof.* We have

$$
\begin{aligned}
\mathbb{E}[X^3] - \mathbb{E}[X^2]\mathbb{E}[X] &= \mathbb{E}[(X^2 - \mathbb{E}[X^2])(X - \mathbb{E}[X])] \\
&= \mathbb{E}[(X^2 - \mathbb{E}[X]^2)(X - \mathbb{E}[X])] + \mathbb{E}[(\mathbb{E}[X]^2 - \mathbb{E}[X^2])(X - \mathbb{E}[X])] \\
&= \mathbb{E}[(X + \mathbb{E}[X])(X - \mathbb{E}[X])^2] + 0 \\
&\geq 0,
\end{aligned}
$$

which concludes the proof. $\square$

**Lemma H.2** (Freedman's Inequality). *Let $M, v > 0$ be fixed constants. Let $\{X_i\}_{i=1}^n$ be a stochastic process, $\{\mathcal{G}_i\}_i$ be a sequence of $\sigma$-fields, and $X_i$ be $\mathcal{G}_i$-measurable, while almost surely*

$$\mathbb{E}[X_i|\mathcal{G}_i] = 0, |X_i| \leq M, \text{ and } \sum_{i=1}^n \mathbb{E}[X_i^2|\mathcal{G}_{i-1}] \leq v.$$

*Then for any $\delta > 0$, with probability at least $1 - \delta$, it holds that*

$$\sum_{i=1}^n X_i \leq \sqrt{2v \log(1/\delta)} + \frac{2}{3} M \log(1/\delta).$$

**Lemma H.3** (Martingale Exponential Inequalities). *Consider a sequence of random functions $\zeta_1(\mathcal{Z}_1), \ldots, \zeta_t(\mathcal{Z}_t), \ldots$ with respect to filtration $\{\mathcal{F}_t\}$. We have for any $\delta \in (0, 1)$ and $\lambda > 0$:*

$$\mathbb{P}\left[\exists n > 0 : -\sum_{i=1}^n \zeta_i \geq \frac{\log(1/\delta)}{\lambda} + \frac{1}{\lambda}\sum_{i=1}^n \log \mathbb{E}_{Z_i^{(y)}} \exp(-\lambda\zeta_i)\right] \leq \delta,$$

*where $Z_t = (Z_t^{(x)}, Z_t^{(y)})$ and $\mathcal{Z}_t = (Z_1, \ldots, Z_t)$.*

**Lemma H.4** (Multiplicative Chernoff Bounds). *Assume that $X \in [0, 1]$ with $\mathbb{E}X = \mu$. Then for all $\epsilon > 0$,*

$$\mathbb{P}\left(\bar{X}_n \geq (1 + \epsilon)\mu\right) \leq \exp\left[\frac{-2n\mu\epsilon^2}{2 + \epsilon}\right]$$

$$\mathbb{P}\left(\bar{X}_n \leq (1 - \epsilon)\mu\right) \leq \exp\left[\frac{-2n\mu\epsilon^2}{2}\right].$$

*Moreover, for $t > 0$, we have*

$$\mathbb{P}\left(\bar{X}_n \geq \mu + \sqrt{\frac{2\mu t}{n}} + \frac{t}{3n}\right) \leq \exp(-t).$$

*Proof.* Refer to the proof of Corollary 2.18 in Zhang (2023).

*Remark* H.5. The multiplicative chernoff bounds (Lemma H.4) can be expressed as follows. With probability at least $1 - \delta$:

$$\mu \le \bar{X}_n + \sqrt{\frac{2\mu \ln(1/\delta)}{n}}.$$

It implies that for any $\gamma \in (0, 1)$:

$$\bar{X}_n \ge (1 - \gamma)\mu - \frac{\ln(1/\delta)}{2\gamma n}.$$

$\square$

**Lemma H.6.** *Suppose $a, b \ge 0$. If $x^2 \le a + b \cdot x$, then $x^2 \le 2b^2 + 2a$.*

*Proof.* By solving the root of quadratic polynomial $q(x) := x^2 - b \cdot x - a$, we obtain $\max\{x_1, x_2\} = (b + \sqrt{b^2 + 4a})/2$. Hence, we have $x \le (b + \sqrt{b^2 + 4a})/2$ provided that $q(x) \le 0$. Then we further have

$$x^2 \le \frac{1}{4}\left(b + \sqrt{b^2 + 4a}\right)^2 \le \frac{1}{4} \cdot 2\left(b^2 + b^2 + 4a\right) \le 2b^2 + 2a. \tag{20}$$

$\square$

**Lemma H.7.** *Let $X$ be a random variable and $0 \le X \le M$, we have*

$$\mathrm{Var}(X) \le M\mathbb{E}(X).$$

*Proof.* From Bhatia-Davis inequality, we have

$$\mathrm{Var}(X) \le (M - \mathbb{E}(X))\mathbb{E}(X) \le M\mathbb{E}(X).$$

$\square$

**Lemma H.8** (Chi-squared distribution's tail bound). *If $\chi_d^2$ is a $d$-dimensional chi-square distribution, we have the following estimation for the tail bound.*

$$\mathbb{P}\{\chi_d^2 > (1 + \varepsilon)\, d\} \le \exp\left\{-\frac{d}{2}(\varepsilon - \log(1 + \varepsilon))\right\}, \tag{21}$$

*Proof.* Consider the moment-generating function of the distribution $\chi_d^2$

$$M_{\chi_d^2}(t) = (1 - 2t)^{-\frac{d}{2}}, \quad \text{for any } t < \frac{1}{2}.$$

By the Markov's inequality, for any $t > 0$, we have

$$\mathbb{P}\{\chi_d^2 > (1 + \varepsilon)\, d\} \le \exp\{-t(1 + \varepsilon)d\} \cdot M_{\chi_d^2}(t) = \exp\{-t(1 + \varepsilon)d\} \cdot (1 - 2t)^{-\frac{d}{2}}$$

for any $t < \frac{1}{2}$. We choose $t$ to minimize the right-hand side: $\exp\{-t(1 + \varepsilon)d\} \cdot (1 - 2t)^{-\frac{d}{2}}$ and that is $t = \varepsilon/2(1 + \varepsilon)$. Substituting $t$ back into the inequality, we get the desired bound. $\square$

**Lemma H.9** (Online-to-batch conversion). *If an algorithm has a sublinear regret of $c^\dagger \cdot \log T$, then the algorithm finds an $\epsilon$-optimal policy with at most $\widetilde{\Theta}(c^\dagger/\epsilon)$ samples, where $\widetilde{\Theta}$ omit logarithmic terms of $c^\dagger/\epsilon$. Here $c^\dagger$ is a problem-dependent constant.*

*Proof.* We denote the policy sequence as $\{\pi^1, \cdots, \pi^T\}$. Then, by definition of regret, we know

$$\mathrm{Regret}(T) = T \cdot V_1^*(x_1) - \sum_{t=1}^{T} V_1^{\pi^t}(x_1)$$

$$\le c^\dagger \cdot \log T.$$

We consider the uniform policy $\tilde{\pi} := \text{Uniform}(\pi^1, \cdots, \pi^T)$. It follows that

$$V_1^*(x_1) - V_1^{\tilde{\pi}}(x_1) = V_1^*(x_1) - \frac{1}{T}\sum_{t=1}^T V_1^{\pi^t}(x_1) \leq c^\dagger \cdot \frac{\log T}{T}.$$

It suffices to prove that

$$c^\dagger \cdot \frac{\log T}{T} \leq \epsilon,$$

which is equivalent to solving

$$T \leq \exp(T \cdot \epsilon / c^\dagger).$$

By using the Lambert W function[1], we can prove that

$$T \geq \frac{W(1)c^\dagger}{\epsilon},$$

where $W(1) \geq \log(1/\epsilon) - \log\log(1/\epsilon)$. $\qquad\square$

---

[1] https://en.wikipedia.org/wiki/Lambert_W_function

