# OpenReview forum: "$f$-Divergence Regularized RLHF: Two Tales of Sampling and Unified Analyses"
_ICML.cc/2026/Conference — ICML 2026 regular_

### Official Review · Reviewer_BRp2 · 2026-03-03

**Soundness:** 2
**Presentation:** 2
**Significance:** 3
**Originality:** 3
**Overall Recommendation:** 2
**Confidence:** 4

**Summary:**

This paper considers RLHF with $f$-divergence regularized objective, which is formulated as $f$-divergence-regularized contextual dueling bandit. Under this problem formulation, the paper proposes two algorithm for this setup. The first algorithm is based on commonly used optimistic exploration, whose regret as proved in this paper, is upper bounded by $\eta\mathcal{C}\log(T\mathcal{N})d(T)$. The paper further proposes an sampling-based algorithm, which bypasses the construction of exploration bonus. Theoretical analysis shows that the algorithm admits a sample complexity $O(1/\epsilon)$ asymptotically. The paper also include empirical results that verified the theoretical findings

**Compliance With Llm Reviewing Policy:**

Affirmed.

**Final Justification:**

This paper studies RLHF under $f$-divergence regularization. The paper has several merits, namely 1) an unified analysis of contextual bandit under $f$-divergence with general $f$; and 2) A novel sampling algorithm based on the observation that $h'$ connects the estimation error and policy learning error

However, in the review, two major weaknesses are identified:

1. The validity of equation in line 747: The authors do not provide sufficient argument to support the validity of this equation during the rebuttal.

2. Computational inefficiency of Algorithm 2: While it is argued in the rebuttal that algorithm 2 can be made sufficient via some approximation, the rationale behind this approximation is not adequately justified. Therefore, whether this algorithm is superior to bonus-based algorithm remains unclear.

Given these unresolved concerns, I would like to maintain my evaluation.

**Key Questions For Authors:**

Questions for the authors are listed as follows

1. The bound in Theorem 4.4 scales with the quantity $||\bar{T}\_{\theta} + Z^{+}\_{\theta} Z^{-}\_{\theta}\bar{T}\_{\theta}||_{\infty}$. How does this quantity scale with problem parameters (e.g., $A$, $d$, etc.,)?

2. Currently, the bound in Theorem 4.4 only guarantees asymptotic behaviors. Could this be improved to anytime guarantee?

**Limitations:**

See weakness section.

**Strengths And Weaknesses:**

The strengths and weaknesses of this paper are listed as follows

Strengths:

1. This paper provide a unified analysis of $f$-divergence regularized (dueling) bandit, which strictly extends the previous results

2. This paper provides an interesting understanding of the function $h$, whose derivative connect the error of reward estimation and the sub-optimality gap.

3. The proposed Algorithm 2, based on the observation above, leverage a mixture-of-policy sampling strategy for exploration.

4. Theorem 4.3, which serves as the technical foundation of Theorem 4.4, looks like a new result.

Weaknesses:

1. Technical Flaw: The equation mark at line 747 actually requires a more fine-grained analysis. Currently, at line 745, the summation is take over the conditional expectation of the bonus terms, i.e., $\mathbb{E}_{a \sim \pi_t}U(a)^2$. However, the definition of eluder dimension requires the sum to be the exact realized trajectories. That is saying, there is a discrepancy of $\sum\_t [U(a\_t)^2 - \mathbb{E}\_{a \sim \pi\_t}U(a)^2]$ between lline 745 and line 747.

2. Assumption on $f$: This paper requires $f$ to satisfy $0 \notin \text{dom}(f')$, which is a bit constrainted since this requirement excludes a large amount of $f$, for example the $\chi^2$-divergence considered in Algorithm 3 of [1], where $f(x) = x^2/(1+\eta x)$. The paper should clearly state this constraint as an assumption and discuss this limitation, for example replacing Table 1 with Table 2.

3. Computational Inefficiency: Line 6 of Algorithm 2 requries minimizing the function $L(\theta)$ for $\theta$. However, the term $\omega(x)$ depends on $\theta$ by $\bar{T}\_{\theta} + Z^{+}\_{\theta} Z^{-}\_{\theta}\bar{T}\_{\theta}$, which might be highly non-convex and making Algorithm 2 itself computationally inefficient.

4. Ambiguities and Typos:

    a. In Proposition 4.4, the variable $\varepsilon$ is introduced without any prelude. Does Proposition 4.4 holds for all $\varepsilon$ or some specific $\varepsilon$?

    b. In Proposition 4.4, $n$ below limsup should be $T$. Also, $C$ in the definition of $C_0$ is also undefined. $\hat \pi$ is also undefined.

    c. Algorithm 2 should ended with something like "ensure $\hat \pi$".

    d. At the right column of line 217 and right column of line 225, $C$ should be $\mathcal{C}$

---

[1] Correcting the Mythos of KL-Regularization: Direct Alignment without Overoptimization via χ2-Preference Optimization, ICLR 2025

---

> ### Author Rebuttal · Authors · 2026-03-31
>
> Thanks for your comment. Please find our responses to each question below.
>
> ----
>
> **Weakness 1**: The equation mark at line 747 actually requires a more fine-grained analysis.
>
> **Answer 1**: The definition of eluder dimension $d$ is
> $$
>     d(T)=\sup_{x_{1:T},a_{1:T}}\sum_t U(x_t,a_t^1,a_t^2;R,D_{t-1}).
> $$
> And taking the expectation will be less than the supremum.
>
> ----
>
> **Weakness 2**: This paper requires $f$ to satisfy $0\notin \text{dom}(f')$, which is a bit constrainted since this requirement excludes a large amount of $f$.
>
> **Answer 2**: Thanks for the comment. We agree that the condition excludes some choice of divergence. But in the class of $0\notin \text{dom}(f')$, it also contains a wide range of famous divergences, such as reverse KL and forward KL divergence. We will elaborate on this constraint in the revision and replace Table 1 with Table 2 or some more explicit table.
>
> ----
>
> **Weakness 3**: Computational Inefficiency
>
> **Answer 3**: Thanks for the comment. The proposed loss in Algorithm 2 is the ideal case and fits the theoretical framework. In practice, we can take all $\omega(x)=1$, and this will be easier to handle. We will provide more experiments for this on the revision.
>
> ----
>
> **Weakness 4**: Ambiguities and Typos
>
> **Answer 4**: Thanks for pointing those out. We will carefully revise all of them and double-check all pages for ambiguities and typos.
>
> ----
>
> **Question 1**: The bound in Theorem 4.4 scales with the quantity $||\overline T+Z^+Z^-\overline T||_\infty$. How does this quantity scale with problem parameters.
>
> **Answer 5**: We can rewrite $Z^+$ as
> $$
>     Z^+(x)=\mathbb E_{a\sim \pi_\theta'}\exp(r_\theta(x,a)),
> $$
> and it will not scale a lot with $\mathcal{A}$ and it can be bounded by $\exp(R)$ where $r\in[0,R]$. For $Z^-$, we can handle it similarly to $Z^+$. For $\overline{T}$, we can bound it as
>
> $$
> \overline{T}(x)=\sum_a\pi_0(a|x) h'(\eta(r_\theta(x,a)-\lambda_{\theta}(x)))\leq C(f,R_\Theta,\eta)\sum\pi_0(a|x) h(\eta(r_\theta(x,a)-\lambda_{\theta}(x)))=C(f,R_\Theta,\eta).
> $$
>
> By the above bounds, the constant can be bounded by $C(f,R_\Theta,\eta)$ and the reward region $R$.
>
> ----
>
> **Question 2**: Currently, the bound in Theorem 4.4 only guarantees asymptotic behaviors. Could this be improved to anytime guarantee?
>
> **Answer 6**: Thanks for the comments. We first would like to restate that our contributions are: proposing two exploration strategies for the $f$-divergence regularized RLHF and, novelly, using derivative-based strategies to handle the exploration, which provides new insight and benefit computation. For an anytime guarantee, which is also a critical improvement, we will keep working on that as future work.

---

> > ### Author Rebuttal · Reviewer_BRp2 · 2026-04-03
> >
> > I thank the authors for the responses. However, some of my concerns remains:
> >
> > 1. The equation mark at line 747: Let's consider the chain of inequalities here. The goal is to upper bound the random variable $\text{Regret}(T)$. If we omit the dependency on $\eta$, $\mathcal{C}$ and $\beta$, then line 745 upper bounds $\text{Regret}(T)$ by $\sum\_{t=1}^{T}\mathbb{E}\_{a \sim \pi_t}\min \\{1, U(x, a)^2\\}$ . It worth noting that for $t$, the corresponding term in RHS of line 745 is only the conditional expectation up to the filtration till $\pi_t$ (which are still random variables for $t \geq 2$. Therefore, it does not equal to taking expectation over the distribution trajectory (which is a constant number). If, as claimed in the rebuttal, taking expectation over the trajectory, then the RHS indeed becomes $\mathbb{E}\_{x_{1:T}, a_{1:T}}[\sum\_{t=1}^T U(x_t, a_t)^2]$, which, I agree, can be bounded by taking the maximum over all possible trajectories and obtain the upper bound $d_{\text{RF}}$. However, the expectation is taken on both side, which means that $\mathbb{E}\_{x_{1:T}, a_{1:T}}[\sum\_{t=1}^T U(x_t, a_t)^2]$ can only upper bound $\mathbb{E}[\text{Regret}(T)]$. Therefore, line 747 can only establish an upper bound on the expectation of regret, which, conflicts with the high probability statement in Theorem C.3.
> >
> > 2. Computational Inefficiency: I thank the author for the clarification. However, my concern is more about the motivation, or potential benefits of Algorithm 2. At the beginning of Section 4, the paper motivates Algorithm 2 by arguing that bonus-based algorithm involves generally non-trivial computation of the bonus function and claim Algorithm 2 somewhat bypass this. However, this benefit does not hold since Algorithm 2 is also computationally inefficient: It bypasses the computation of the bonus, yet introduces computation of the hard-to-optimize $\theta$. While it might be true that $\omega(x)$ can be approximated, this does not constitute a benefit over exploration bonus based algorithm, since exploration bonus can also be somehow approximated. The paper should provide a stronger motivation for Algorithm 2 given the standing out issue in  computational efficiency.
> >
> > Given the remaining concerns, I would like to keep my current evaluation.
> >
> > ---
> >
> > Update: I just saw the authors' clarifications. I would like to make some further elaboration:
> >
> > 1. Line 747. Let me further elaborate the issue here. For simplicity, we drop the context $x$, so that a trajectory is given by $a_{1:T}$. Let's also omit $\eta$, $\mathcal{C}$ and $\beta$, and the $\mathcal{R}$, $\xi$ and $\min\\{1, \cdot\\}$ in the expression of eluder dimension. That is, after the simplification, the eluder dimension is defined as $d = \sup_{a_{1:T}}\sum_{t=1}^T U(a_t, \mathcal{D_{t-1}})^2$. After this notational simplification, the RHS of line 745 looks like $\sum\_{t=1}^{T}\mathbb{E}\_{a \sim \pi_t}U(a, \mathcal{D}_{t-1})^2$.
> > > We want to mention that no matter what the action is chosen, it will be less than the supremum (given that the policy
> >  is exactly where we sample the action).
> >
> >     If my understanding is correct, the authors' argument here is that we substitute the $\mathbb{E}\_{a \sim \pi_t}$ with $\max_{a \in \mathcal{A}}$ (Please correct me if you do not mean by this). This leads to RHS of 745 being upper bounded by $\sum\_{t=1}^{T}\max_{a \in \mathcal{A}}U(a, \mathcal{D}\_{t-1})^2$, or $\\sum\_{t=1}^{T} U(a'\_t, \mathcal{D}\_{t-1})^2$, where $a'\_t = \\mathrm{argmax}\_{a \in \mathcal{A}} U(a, \mathcal{D}\_{t-1})$. However, if we compare this with the definition of eluder dimension $d = \sup\_{a_{1:T}}\sum\_{t=1}^T U(a_t, \mathcal{D\_{t-1}})^2$, we see a mismatch of the position of $\sup$.
> >
> >     For each possible trajectory incorporated by the $\sup\_{a\_{1:T}}$ in the definition of eluder dimension, despite that there is no restriction on each $a_t$, the definition of the dataset $\mathcal{D}$ gives that $\mathcal{D}\_t = \mathcal{D}_{t-1} : \\{a_t\\}$. That's saying, for any two consecutive summands, $U(a_t, \mathcal{D\_{t-1}})^2$ and $U(a\_{t+1}, \mathcal{D\_t})^2$, they must admits $\mathcal{D}\_t = \mathcal{D}\_{t-1} : \\{a_t\\}$.
> >
> >     However, in the upper bound we just derived for RHS of line 745, $\\sum\_{t=1}^{T} U(a'\_t, \mathcal{D}\_{t-1})^2$, it is generally not guaranteed that $\mathcal{D}\_t = \mathcal{D}\_{t-1} : \\{a'\_t\\}$, since this requires exactly $a'\_t = a_t$. Therefore, $\\sum\_{t=1}^{T} U(a'\_t, \mathcal{D}\_{t-1})^2$ cannot be upper bounded by $d=\sup\_{a_{1:T}}\sum\_{t=1}^T U(a_t, \mathcal{D\_{t-1}})^2$ in general. Could the author show how to overcome this challenge?
> >
> >
> > 2. About Algorithm 2: I thank the author for the response. However, the rebuttal should provide substantial justification that $\omega=1$ is a reasonable choice. Simply saying setting $\omega=1$ in practice makes it efficient is unfair, since one can also claim exploration bonus is efficient by setting the confidence radius to 0.

---

> > > ### Author Response · Authors · 2026-04-04
> > >
> > > - First, we would like to emphasize the key idea behind Algorithm 2: we use the derivative to replace the hard-to-compute uncertainty bonus, which, to the best of our knowledge, is the first use of such a strategy. This corresponds to the exploration part of the algorithm, which is distinct from the MLE part you mentioned. It explores efficiently and avoids computing the uncertainty bonus. For optimizing $\theta$, we can still set $\omega=1$ in practice. Under this choice, Algorithm 2 achieves substantially better computational efficiency.
> > >
> > > - We want to mention that no matter what the action is chosen, it will be less than the supremum (given that the policy $\pi_t$ is exactly where we sample the action). We agree that $\pi_t$ is also a random variable, but we don't need to consider $\mathbb E_{x_{1:T},a_{1:T}}$.
> > >
> > > We hope that these two points help resolve the two concerns.

---

### Official Review · Reviewer_YhRg · 2026-03-07

**Soundness:** 3
**Presentation:** 3
**Significance:** 2
**Originality:** 2
**Overall Recommendation:** 4
**Confidence:** 3

**Summary:**

This paper proposes two exploration strategies for $f$-divergence-regularized RLHF: the first is optimism-based, while the second is derivative-based. The paper analyze theoretically the performance of the two methods and establishes an upper bound on the regret of $O(\log(T))$ for the optimism-based exploration algorithm, as well as a bound of $O(1/T)$ on the suboptimality gap for derivative-based exploration.

**Compliance With Llm Reviewing Policy:**

Affirmed.

**Final Justification:**

The authors have successfully addressed my concerns regarding the theoretical part of the paper and improved my understanding of their contributions. However, they did not provide larger-scale experiments. In light of this, I am increasing my score to 4.

**Key Questions For Authors:**

See the weakness section.

**Limitations:**

Yes

**Strengths And Weaknesses:**

## Strenghs:

1) The paper addresses a timely problem of analyzing RLHF under different choices of regularization and proposes two methods to solve this problem.
2) The paper proposes a unifying analysis of the two proposed methods for a general class of f-divergences.
3) The paper is clear and well-written.

## Weaknesses:

I have two main concerns regarding this paper.

1) **First main concern**: There is a strong similarity between several key concepts and results in this paper and those in [1], but these connections are not acknowledged. These similarities undermine the novelty of the results. I detail these connections below:

a) Proposition 2.3, which provides the expression of the optimal policy of the \(f\)-regularized, and the choice of the parametrization (line 133) have already been proposed, respectively, in Section 3 of [1].

b) The convergence rates obtained by both methods are either asymptotic or non-explicit, which are weaker than the non-asymptotic and explicit rates derived in [1]. Additionally, both methods are computationally more costly than the method proposed in [1], which is a simple policy gradient method.

2) **Second main concern**: The bound provided on the regret of algorithm 1 is proportional to

$$ d(R_{\theta}, \xi, T) := \sup_{x_{1:T},a_{1:t}^{1}, a_{1:t}^{2 }} \sum_{t \in [T]} ...$$

It is absolutely not clear that this term does not scale linearly in $T$ as it is a sum over $T$ terms. The authors should properly bound this term to ensure that the optimism-based algorithm achieves a logarithmic regret as claimed in the introduction.

3) The experiments are relatively small-scale compared to those in other RLHF papers. More realistic experiments would further strengthen the message regarding the benefits of the proposed derivative-based exploration method.

[1] Labbi et al, Beyond Softmax and Entropy: Convergence Rates of Policy Gradients with f-SoftArgmax Parameterization \& Coupled Regularization, The Fourteenth International Conference on Learning Representations, 2026.

---

> ### Author Rebuttal · Authors · 2026-03-31
>
> Thanks for your comment. Please find our responses to each question below.
>
> ----
>
>
> **Weakness 1**: There is a strong similarity between several key concepts and results in this paper and those in [1].
>
> **Answer 1**: Thanks for the comment.
>
> - The first point you mention is Proposition 2.2, which we take as a known result for f-regularized RLHF, and we didn't include that for the contribution of this paper.
>
> - As the second similarity you mentioned, paper [1] uses a stochastic policy-gradient / REINFORCE-type update algorithm. For our paper, the two proposed algorithms are more like model-based optimistic planning algorithms. The two key contributions of our paper are: 1) we can achieve logarithmic regret with a general
> $f$-divergence, other than just KL divergence, in optimistic planning algorithms; 2) we provide a novel theoretical insight applicable to any $f$-divergence: the derivative related $f$ can be interpreted as part of the uncertainty.
>
> ----
>
> **Weakness 2**: The term $d(R_{\theta}, \xi, T) := \sup_{x_{1:T},a_{1:T}^{1}, a_{1:T}^{2 }} \sum_{t \in [T]} ...$ is absolutely not clear that this term does not scale linearly in $T$ as it is a sum over $T$ terms.
>
> **Answer 2**: Thanks for the comment. The eluder dimension $d$ is related to the structure of the function class, which will affect the training performance. If the reward function class is assumed as linear, $\mathcal{R}=${$\theta^\top \phi(\cdot,
> \cdot):\theta\in R^d,\| \theta |_2\leq B$}.
>
> Let the  covariance matrix $\Sigma_t=\sum_t \lambda / B \cdot I+(\phi(x_i,a_i^1)-\phi(x_i,a_i^2))(\phi(x_i,a_i^1)-\phi(x_i,a_i^2))^T$. Then, the uncertainty factor can be bounded as
> $$
>     U(x_i,a_i^1,a_i^2;D)= \sup_{\theta_1,\theta_2}\frac{(\theta_1-\theta_2)^\top(\phi(x_i,a_i^1)-\phi(x_i,a_i^2))}{\sqrt{\lambda+\sum_{t=1}^{i}((\theta_1-\theta_2)^\top(\phi(x_i,a_i^1)-\phi(x_i,a_i^2)))^2}}
> $$
>
> $$
>     \leq \sup_{\theta_1,\theta_2}\frac{(\theta_1-\theta_2)^\top(\phi(x_i,a_i^1)-\phi(x_i,a_i^2))}{\sqrt{(\theta_1-\theta_2)^\top\Sigma_i(\theta_1-\theta_2)}}
> $$
>
> $$
>     \leq ||\phi(x_i,a_i^1)-\phi(x_i,a_i^2)||_{\Sigma_i}.
> $$
>
> And $d(R,\xi,T)\leq\sum ||\phi(x_i,a_i^1)-\phi(x_i,a_i^2)||_{\Sigma_i}$ can be bounded as $O(\log T)$ using classical elliptical potential lemma.
>
> ----
>
>
> **Weakness 3**: The experiments are relatively small-scale compared to those in other RLHF papers.
>
> **Answer 3**: Thanks for the comment. We will add some large-scale experiments to verify our algorithms and theoretical findings.

---

> > ### Author Rebuttal · Reviewer_YhRg · 2026-04-03
> >
> > I thank the authors for their rebuttal. I acknowledge that the first two weaknesses I pointed out have been addressed. I also strongly encourage the authors to clarify in the paper that $d(R_{\theta}, \xi, T)$ can be bounded by $\log(T)$ when the function class is linear. The rebuttal helped me better understand the authors' theoretical contributions and results.
> >
> > That said, the experiments are still conducted at a very small scale and do not yet make it possible to assess whether using alternative $f$-divergences for RLHF is practically interesting. I will therefore increase my score to $4$, as I find the theoretical results interesting. If the authors are able to provide more convincing experimental evidence before the end of the rebuttal, I would be willing to increase my score further.

---

> > > ### Author Response · Authors · 2026-04-04
> > >
> > > Thank you for the helpful suggestion. We will add more discussion on how $d(R_{\theta}, \xi, T)$ scales with $T$, especially in the linear setting. We thank you for recognizing the theoretical contribution. We are also currently conducting large-scale experiments and will share the results as soon as they become available; these results will be included in the revision.

---

### Official Review · Reviewer_B1Cg · 2026-03-08

**Soundness:** 3
**Presentation:** 3
**Significance:** 3
**Originality:** 3
**Overall Recommendation:** 4
**Confidence:** 1

**Summary:**

To fill the research gap that there is not a unified theoretical understanding of general f-divergence regularization, this paper develops a comprehensive theoretical framework for online RLHF with an f-divergence regularized objective. Specifically, they propose two exploration strategies: optimism-based exploration and derivative-based exploration. Besides, they establish regret upper bounds over a time horizon T for both algorithms.

**Compliance With Llm Reviewing Policy:**

Affirmed.

**Final Justification:**

The paper studies a meaningful problem, and its main strength is the strong theoretical contribution. The analysis is solid. Overall, I view the paper positively due to its theoretical depth, and acceptable empirical validation.

**Key Questions For Authors:**

(1)	For a RLHF paper, the current synthetic linear scenario is simple. Is it possible for authors to add at least one real LLM experiment to validate your method?
(2)	Can authors clarify whether some f choices yield a better dependence under realistic parameter regimes?
(3)	Can authors discuss the impact of the divergence choice f of h?

**Limitations:**

Yes.

**Strengths And Weaknesses:**

Strengths:
(1)	This paper is well written and makes a theoretical contribution to understanding the online f-divergence regularized RLHF framework.
(2)	This paper offers several interesting insights such as interpreting derivatives as a form of uncertainty which could benefit future research.
(3)	The entire framework is written for a general f-divergence, and covers KL regularization as one specific choice of f.

Weaknesses:
(1)	The feasibility of the proposed algorithms in LLM scale settings is not well demonstrated.
(2)	Current empirical study is simple.

---

> ### Author Rebuttal · Authors · 2026-03-31
>
> Thanks for your comment. Please find our responses to each question below.
>
> ----
>
> **Weakness 1**: The feasibility of the proposed algorithms in LLM scale settings is not well demonstrated.
>
> **Weakness 2**: Current empirical study is simple.
>
> **Question 1**: For a RLHF paper, the current synthetic linear scenario is simple. Is it possible for authors to add at least one real LLM experiment to validate your method?
>
> **Answer 1**: Thanks for pointing that out. We will add some large-scale experiments, including LLM-level experiments, to demonstrate our algorithms and theoretical findings.
>
> ----
>
> **Question 2**:  Can authors clarify whether some f choices yield a better dependence under realistic parameter regimes?
>
> **Answer 2**: First, we want to clarify that the different values of constants $\mathcal{C}$ and $\mathcal{M}$ in the bound guarantee of the theoretical upper bound, which cannot directly transfer to the practical performance. But we have some examples, such as chi-squared mixed KL ($f=(x-1)^2+x\log x$), the constants $\mathcal{C}$ and $\mathcal{M}$ are smaller than those in the KL divergence case, and [1] shows that chi-squared mixed KL is more robust to overoptimization during training. Also, our experiments show that in the selected example, the smaller $\mathcal{C}$ and $\mathcal{M}$ have a better performance.
> The connection between the constant $\mathcal{C},\mathcal{M}$ and the benefit of choice $f$ needs further investigation and will be our future work.
>
> ----
>
> **Question 3**:  Can authors discuss the impact of the divergence choice f of h?
>
> **Answer 3**: Yes. The $h$ is defined as the inverse function of the derivative of $f$. When the derivative of $f$ increases faster, $h$ will be the opposite, i.e., increases more slowly. For a choice of $f$-divergence, $\pi(a|x)\propto\pi_0(a|x)h(r(x,a)+\lambda)$ where $h$ is on the close-form solution.
>
>
> [1] Huang, A., Zhan, W., Xie, T., Lee, J. D., Sun, W., Krishnamurthy, A., and Foster, D. J. Correcting the mythos of kl-regularization: Direct alignment without overoptimization via chi-squared preference optimization, 2025. URL
> https://arxiv.org/abs/2407.13399.

---

> > ### Author Rebuttal · Reviewer_B1Cg · 2026-04-01
> >
> > Thanks for the rebuttal. The authors clarify several theoretical points. I will maintain my score.

---

### Official Review · Reviewer_zqmF · 2026-03-13

**Soundness:** 2
**Presentation:** 3
**Significance:** 2
**Originality:** 2
**Overall Recommendation:** 2
**Confidence:** 4

**Summary:**

This submission presents derives the first logarithmic regret upper bound for learning contextual (dueling) bandi against general $f$-divergence-regularized regret under general function approximation with certain regularity conditions on $f$, which is achieved by an optimism-based algorithm, extending the results in Zhao et al. 2025 for reverse KL.

**Compliance With Llm Reviewing Policy:**

Affirmed.

**Final Justification:**

> I tend to keep my evaluation for the following reasons, nearly all of which were already mentioned in either the review or the rebuttal acknowledgement.

- The derivation of the key lemma (Lemma C.6, Value Decomposition Bound) requires (from Line 840 to Line 841) the $C(f, \mathcal{R}\_\Theta, \eta)$ to be defined by taking $\sup$ over the **convex hull** of the function class $\mathcal{R}_\Theta$, which is an artifact introduced in this submission and such an artifact is absent from nearly all previous related works cited in this paper.
- Line 368 indicates that the newly introduced **function-class-dependent** notion $C(f, \mathcal{R}_\Theta, \eta)$ might be potentially unbounded.
- The hand-wavy derivation on Line 747 overlooks the fact that the notion of $U_\mathtt{RF}$ depends on the on-policy history up to step $t-1$, which, means that the authors are essentially bounding the sum of $\sup$ from above by the $\sup$ of sum; and thus, not justified.
- The analysis of the derivative-based exploration is based on **asymptotic normality**, which is not very appropriate for post-training because the size of the dataset for post-training is really super small compared with that for pre-training.

**Key Questions For Authors:**

1. What is the relationship among the $n$ in Proposition 4.4, the $T$ in Proposition 4.4, and the $t$ on **Line 6 of Algorithm 2**?
   - Furthermore, by letting $n \to \infty$ in Proposition 4.4, which quantity stated in the algorithm block of Algorithm 2 goes to infinity?
2. In the last sentence on the LHS of Page 7, whether the "looser" means the "nature of this divergence" is "looser" or the "proof technique" is "looser" for this divergence?
   - This is not a question directly related to the mathematical evaluation of this paper, but it may help the broader audience understand the results.
3. Open-ended question: Do the authors consider $0 \notin \mathrm{dom}(f')$ an essential or fundamental structure for $f$-divergence regularized objectives?

**Limitations:**

Yes.

**Strengths And Weaknesses:**

### Strengths

- The optimism-based algorithm is intuitively is easy to follow
- This is the first work generalizing the logarithmic regret gaurantee for reverse KL to general $f$-divergences under certain regularity conditions of $f$.

### Weaknesses

- From Line 1530 to Line 1556: More than **20 lines of verbatim __copy-paste__ level text overlap** between the Lemma H.9 (plus its proof) and the last page of [4, Lemma D.2 (and its proof)]
- From Line 1441 to 1453: More than **10 lines of verbatim copy-paste level text overlap** between the proof of Lemma H.8 and [3, equation (45)]. In particulart, such a "convariance"-type structure, though mathematically straightforward, is not identified for the first time in this submission in the analysis of value-difference-type lemmas of *divergence-regularized* decision-making objectives. In detail, [3, equation (45)] and [4, the 3rd equation block from bottom up on page 14] have already identified such a structure (and its typical use case on Line 872 of this submission) in the "fast rate"-type analysis of learning against divergence-regularized objectives. According to [4, page 14], this "covaraince"-type structure in the regularized performance difference lemma (aka value difference lemma) has already been known, at the latest, in [5, page 9].
- The last inequality in the first equation block on Page 16, given the current Assumption 3.1, is **problematic**. In detail, $r'$ in the $T_{r'}(x,a)$ is certain **convex combination** between $r$ and $R$, therefore, to bound it from above using $C(f, R_{\Theta}, \eta)$, (according to the definition of $C(f, R_{\Theta}, \eta)$) the authors essentially need $R_{\Theta}$ to be **convex**, which is not compatible with $|R_{\Theta}| < \infty$ in Assumption 3.1 (unless $R_{\Theta}$ degenerates).
  - Furthermore, most previous works in this line (on the fast rate or logarithmic regret of learning against regularized objectives) generally do not need to assume $R_{\Theta}$ to be convex (in both online and offline settings), e.g., [1], [2], and so on.

References

[1] Aminian, G., Asadi, A. R., Shenfeld, I., & Mroueh, Y. (2025). KL-Regularized RLHF with Multiple Reference Models: Exact Solutions and Sample Complexity. arXiv preprint arXiv:2502.01203.

[2] Wu, Y., Thareja, R., Vepakomma, P., & Orabona, F. (2025). Offline and Online KL-Regularized RLHF under Differential Privacy. arXiv preprint arXiv:2510.13512.

[3] https://arxiv.org/pdf/2510.13060v2

[4] https://openreview.net/pdf?id=6QH9IB53uy

[5] https://arxiv.org/pdf/2502.06051v1

---

> ### Author Rebuttal · Authors · 2026-03-31
>
> Thank you for your comment. We provide our responses to each question below.
>
> ----
>
> **Weakness 1**: From Line 1530 to Line 1556: More than 20 lines of verbatim copy-paste level text overlap between the Lemma H.9 (plus its proof) and the last page of [4, Lemma D.2 (and its proof)]
>
> **Weakness 2**: From Line 1441 to 1453: More than 10 lines of verbatim copy-paste level text overlap between the proof of Lemma H.8 and [3, equation (45)].
>
> **Answer 1**: Thanks for the comment.
> Lemma H.8., H.9. are classical results, and we include the proof in section Auxiliary Lemmas for completeness. We want to restate that the objective we study is a general $f$-divergence regularized objective and doesn't have an explicit formula like the KL-divergence ([3,4,5]), which makes the analysis differ. Our analysis shows that we can achieve logarithmic regret with a general $f$-divergence, other than just KL.
>
> ----
>
> **Weakness 3**: The last inequality in the first equation block on Page 16, given the current Assumption 3.1, is problematic.
>
> **Answer 2**: Thanks for pointing that out. We don't need to assume the reward class is infinite or convex. We will change the definitions of the constants $\mathcal{C},\mathcal{M}$ to the convex hull of the reward functions, and the others will remain unchanged.
>
> ----
>
> **Question 1**: What is the relationship among the $n$ in Proposition 4.4, the $T$ in Proposition 4.4, and the $t$ on Line 6 of Algorithm 2
>
> **Answer 3**: The $n$ in Proposition 4.4 should be the training time, and we will change that in the revision. The $t$ on Line 6 of Algorithm 2 is the $t$ iteration during total time $T$.
>
>
> ----
>
> **Question 2**: In the last sentence on the LHS of Page 7, whether the "looser" means the "nature of this divergence" is "looser" or the "proof technique" is "looser" for this divergence?
>
> **Answer 4**:  First, we want to clarify that the different values of constants $\mathcal{C}$ and $\mathcal{M}$ in the bound guarantee of the theoretical upper bound, which cannot directly transfer to the practical performance. But we have some examples, such as chi-squared mixed KL ($f=(x-1)^2+x\log x$), the constants $\mathcal{C}$ and $\mathcal{M}$ are smaller than those in the KL divergence case, and [6] shows that chi-squared mixed KL is more robust to overoptimization during training. Also, our experiments show that in the selected example, the smaller $\mathcal{C}$ and $\mathcal{M}$ have a better performance.
> The connection between the constant $\mathcal{C},\mathcal{M}$ and the benefit of choice $f$ needs further investigation and will be our future work.
>
> ----
>
> **Question 3**: Open-ended question: Do the authors consider $0\notin\text{dom}(f')$ an essential or fundamental structure for $f$-divergence regularized objectives?
>
> **Answer 5**: Thanks for the question. First, we would like to mention that this constraint already contains a wide range of interesting divergences, such as reverse KL and forward KL divergence. For the other choice of $f$ (i.e. $0\in\text{dom}(f')$), we can still get a similar solution, like Proposition 2.3. by the dual function of $f$. But based on the proposition of dual function will be harder to further analyze and might lead to other results. We will study this more general case in our future work.
>
> [6] Huang, A., Zhan, W., Xie, T., Lee, J. D., Sun, W., Krishnamurthy, A., and Foster, D. J. Correcting the mythos of kl-regularization: Direct alignment without overoptimization via chi-squared preference optimization, 2025. URL
> https://arxiv.org/abs/2407.13399.

---

> > ### Author Rebuttal · Reviewer_zqmF · 2026-04-01
> >
> > Thank you for the reply.
> >
> > Regarding Question 1, I guess (from the authors' answer to this question), that the $n$ in Proposition 4.4 should be replaced by $T$. **With that said**, Proposition 4.4 is an **asymptotic** bound based on **asymptotic normality**. Then why could the authors deduce from this asymptotic result a **non-asymptotic** high-probability $O(1/T)$ guarantee for Algorithm 2 with specific level of confidence probability (in the paragraph right below Proposition 4.4)?
> > - Furthermore, what prevents the authors from deriving a standard non-asymptotic high-probability guarantee for Algorithm 2, and why would we need to resort the **asymptotic normality** here?
> > - Minor follow-up question: is the derivative-based exploration here in Algorithm 2 **similar in spirit** to the "differentiable function approximation" in https://arxiv.org/pdf/2210.00750?
> >
> > Regarding Weakness 3, if the $\mathcal{C}$ and $\mathcal{M}$ are defined based on a $\sup$ over the **convex hull** of the reward function class, that feels like a stronger notion of assumption compared with previous works.
> >
> > **Major follow-up** question: for the proof of Theorem C.3, could the authors elaborate why could we bound $\mathbb{E}_{a\sim \pi_t}(...)$ with the Eluder dimension (the later of which in Definition 3.2 seems to be defined on the **on-policy** trajectory while the former of which seems to be an **averaged on-policy** notion)?
> >
> >
> > Regarding Weaknesses 1 and 2, I did no say these lemmata are mathematically heavy, but these structures (in the context of deriving the fast rate of learning with a bonus term with respect to certain divergence-regularized objectives) are not identified for the first time by the authors, as I mentioned in the initial review.

---

> > > ### Author Response · Authors · 2026-04-01
> > >
> > > Thanks for the follow-up.
> > >
> > > ----
> > >
> > >
> > > **Follow-up for Weakness 3**: We agree that change the definition in the $\mathcal{C}$ and $\mathcal{M}$ to the convex hull will make the notion stronger. But the basic propositions of the value of $\mathcal{C},\mathcal{M}$ will remain the same (Section 4.4). And sometimes, defining the region as a convex hull (which is convex) has some computational benefits.
> > >
> > > ----
> > >
> > > **Follow-up for Question 1**: Thanks for the comment. In the paragraph you mentioned, the result we want to claim is not a non-asymptotic high-probability guarantee. We want to mention that when $T$ is big enough, we have the $O(1/T)$ result.
> > >
> > > - For this algorithm, the most important thing we want to deliver is how we use the derivative, a novel aspect, to perform exploration. The analysis and result, although asymptotic, show that the insight for the derivative-exploration is correct.
> > >
> > > - We think our work is different from the paper you mentioned. First, the derivative of paper (https://arxiv.org/pdf/2210.00750) is the gradient of the estimation of $Q$ function, and they use it by combining with past data to construct pessimism-bonus. Our paper focuses on the $f$-divergence regularized setting and leverages the derivative of the function $f$, which differs from the derivative of the reward function.
> > >
> > > ----
> > >
> > > **Major follow-up question**: For the formula:
> > > $$
> > > \sum_{t=1}^T \mathbb E_{x\sim\rho,a\sim \pi_{t}}\min(1,(U(\xi,x_t,a_t;R_{t-1},D_{t-1}))^2),
> > > $$
> > > where $\pi_t$ is the policy we used to sample the answer $a_t$, the data set $D_T$ we sample is from $\pi_{1:T}$.And the eluder dimension is defined by the supremum of the on-policy trajectory.
> > >
> > > ----
> > >
> > > **Weaknesses 1 and 2**: Thanks for the advice. We will add citations for the two lemmas in the revision.
> > >
> > > ----
> > >
> > > Please do not hesitate to reach out if you have any further concerns or questions.

---

### Decision · Program_Chairs · 2026-04-30

**Decision:**

Accept (regular)

**Comment:**

There was substantial disagreement among the reviewers of this paper.  Some reviewers were concerned with the technical correctness of the work.  Some of this I believe is due to reviewer ignorance of a standard argument involving the elliptical potential in the appendix.  While I agree that it may be better to include more details, line 747 seems correct to me as written.  Other reviewers were satisfied with the authors' rebuttal, although continued to believe that further experiments would be desirable.  After having reviewed the theory myself, I do not believe that it is incorrect.